# Nucleosome conformation dictates the histone code

**Matthew R Marunde[1†], Harrison A Fuchs[2,3†], Jonathan M Burg[1], Irina K Popova[1], Anup Vaidya[1], Nathan W Hall[1], Ellen N Weinzapfel[1], Matthew J Meiners[1], Rachel Watson[1], Zachary B Gillespie[1], Hailey F Taylor[1], Laylo Mukhsinova[1], Ugochi C Onuoha[1], Sarah A Howard[1], Katherine Novitzky[1], Eileen T McAnarney[1], Krzysztof Krajewski[4], Martis W Cowles[1], Marcus A Cheek[1], Zu-Wen Sun[1], Bryan J Venters[1], Michael-C Keogh[1]\*, Catherine A Musselman[2,3]\***

[1]EpiCypher, Durham, United States; [2]Department of Biochemistry, University of Iowa Carver College of Medicine, Aurora, United States; [3]Department of Biochemistry and Molecular Genetics, University of Colorado Anschutz Medical Campus, Aurora, United States; [4]Department of Biochemistry and Biophysics, University of North Carolina at Chapel Hill, Chapel Hill, United States

**\*For correspondence:**
mkeogh@epicypher.com (M-CK);
catherine.musselman@
cuanschutz.edu (CAM)

[†]These authors contributed equally to this work

**Abstract** Histone post-translational modifications (PTMs) play a critical role in chromatin regulation. It has been proposed that these PTMs form localized 'codes' that are read by specialized regions (reader domains) in chromatin-associated proteins (CAPs) to regulate downstream function. Substantial effort has been made to define [CAP: histone PTM] specificities, and thus decipher the histone code and guide epigenetic therapies. However, this has largely been done using the reductive approach of isolated reader domains and histone peptides, which cannot account for any higher-order factors. Here, we show that the [BPTF PHD finger and bromodomain: histone PTM] interaction is dependent on nucleosome context. The tandem reader selectively associates with nucleosomal H3K4me3 and H3K14ac or H3K18ac, a combinatorial engagement that despite being in cis is not predicted by peptides. This in vitro specificity of the BPTF tandem reader for PTM-defined nucleosomes is recapitulated in a cellular context. We propose that regulatable histone tail accessibility and its impact on the binding potential of reader domains necessitates we refine the 'histone code' concept and interrogate it at the nucleosome level.

## Editor's evaluation

The manuscript investigates how the tandem reader domains in BPTF co-recognize two types of modifications present on histone tails, H3K4me3 and H3 acetylation. The authors provide compelling evidence for regulation of such recognition by the conformational restriction of histone tails due to interactions with nucleosomal DNA. The findings contribute valuable new insights into how the nucleosomal context impacts the action of tandem reader domains and should be of much interest to the broader chromatin field.

## Introduction

The eukaryotic genome exists in the form of chromatin, with the basic repeating nucleosome subunit a core-histone octamer (two each of H2A, H2B, H3, and H4) wrapped by ~147 base pairs of DNA (*Figure 1a*; *Luger et al., 1997*). Chromatin organization is critical for regulation of the underlying genome, and is spatially and temporally controlled through development and within somatic cells. A major potential mechanism to modulate chromatin structure is the posttranslational modification

(PTM) of the histone proteins (*Figure 1a*). Globally speaking, particular histone PTMs are correlated with distinct chromatin states (e.g. transcriptional activation/repression, damaged DNA) and/or genomic elements (e.g. gene promoters, transcriptional enhancers, centromeres) (*Wang et al., 2009*; *Zhou et al., 2011*; *Rivera and Ren, 2013*). Importantly, it has been proposed that the PTMs function in diverse combinations, perhaps even forming a 'histone code' (*Strahl and Allis, 2000*; *Gardner et al., 2011*; *Lee et al., 2010*) read by chromatin associated proteins (CAPs) via their various 'reader domains,' thus localizing and/or regulating CAP activity (*Rothbart and Strahl, 2014*; *Andrews et al., 2016a*). However, the dictates of such a code, and the role of reader domains in its interpretation, are hotly debated, as it has been challenging to: determine the PTM pattern(s) read out by tandem domains in vitro, determine whether such patterns are actually being engaged in vivo, and finally determine if this has a biological outcome (*Gardner et al., 2011*; *Rando, 2012*; *Smith and Shilati-fard, 2010*; *Allis and Jenuwein, 2016*). Resolving this situation is critical to define the fundamentals of any histone code, utilize PTM patterns in disease diagnostics, and therapeutically target CAP-PTM associations (*Kelly et al., 2010*; *Ahuja et al., 2016*; *Önder et al., 2015*; *Zaware and Zhou, 2017*).

As a starting point, it is necessary to clearly establish the PTM patterns actually engaged by reader domains. To date, the in vitro specificity of individual readers has primarily been determined with modified histone peptides (*Andrews et al., 2016a*; *Patel and Wang, 2013*; *Musselman et al., 2012*), with the selectivity of grouped domains generally derived from a simple sum of individual reader specificities (*Patel and Wang, 2013*; *Ruthenburg et al., 2007*). However, many enzymes that act on histone tails show altered activity on peptide and nucleosome substrates (*Allali-Hassani et al., 2014*; *Kim et al., 2020*; *Strelow et al., 2016*; *Stützer et al., 2016*; *Marabelli et al., 2019*; *Jain et al., 2023*; *Thomas et al., 2023*; *Wang et al., 2023*). Similarly, multiple reader domains display modified interaction with histone tail PTMs in the nucleosome context (*Morrison et al., 2018*; *Wang and Hayes, 2007*; *Gatchalian et al., 2017*; *Peng et al., 2021*; *Ruthenburg et al., 2011*; *Spangler et al., 2023*). This suggests a wide-ranging impact of the higher-order environment, and undermines the common approach of using positive peptide data to select nucleosomes for further analysis.

Here, we instead take an unbiased approach to examine how nucleosome context could alter histone PTM pattern readout, using the BPTF PHD-BD tandem reader as a model system. We confirm the generally observed decrease in the affinity of each reader for nucleosomal histone tails relative to peptides, but also a nucleosomal restriction in the preferred PTM pattern. Our data suggests this is largely due to the reduced accessibility of histone tails in the nucleosome context, where the tails must be displaced from DNA to enable PTM readout. This alters the binding of individual domains, and the multivalent engagement of tandem domains. We propose the 'histone code' is ultimately defined by a combination of three elements: (1) the PTMs that can be recognized and bound by individual reader domains; (2) accessibility of the modified histone tails in the nucleosome context; and (3) the organization and multivalent binding potential of grouped domains (where the whole is greater than the sum of the parts).

## Results
### BPTF PHD-BD shows restricted and synergistic binding in the nucleosome context
The BPTF subunit is important for chromatin association of the NURF (Nucleosome Remodeling Factor) complex (*Wysocka et al., 2006*; *Li et al., 2006*), and pro-tumorigenic in several malignancies (*Zahid et al., 2021b*). At the BPTF C-terminus is a tandem of reader domains: a PHD finger and bromo-domain (PHD-BD, *Figure 1b*). These are of interest for targeted therapeutics (*Zahid et al., 2021a*), so an understanding of their function has direct application. In the context of histone peptides, the PHD associates with H3 tri-methylated at lysine 4 (H3K4me3) (*Li et al., 2006*), while the BD binds H3 and H4 tails containing acetylated lysines (Kac), with a preference for H4 (*Ruthenburg et al., 2011*; *Li et al., 2006*; *Olson et al., 2020*). While efforts have been made to investigate the recruitment of BPTF PHD-BD to modified nucleosomes, only a limited subset of H3K4me3/H4Kac combinations based on peptide data have been tested, suggesting a preference for [H3K4me3, H4K5acK8acK12acK16ac (hereafter H4tetra[ac])] (*Ruthenburg et al., 2011*) or [H3K4me3, H4K16ac] (*Ruthenburg et al., 2011*; *Nguyen et al., 2014*).

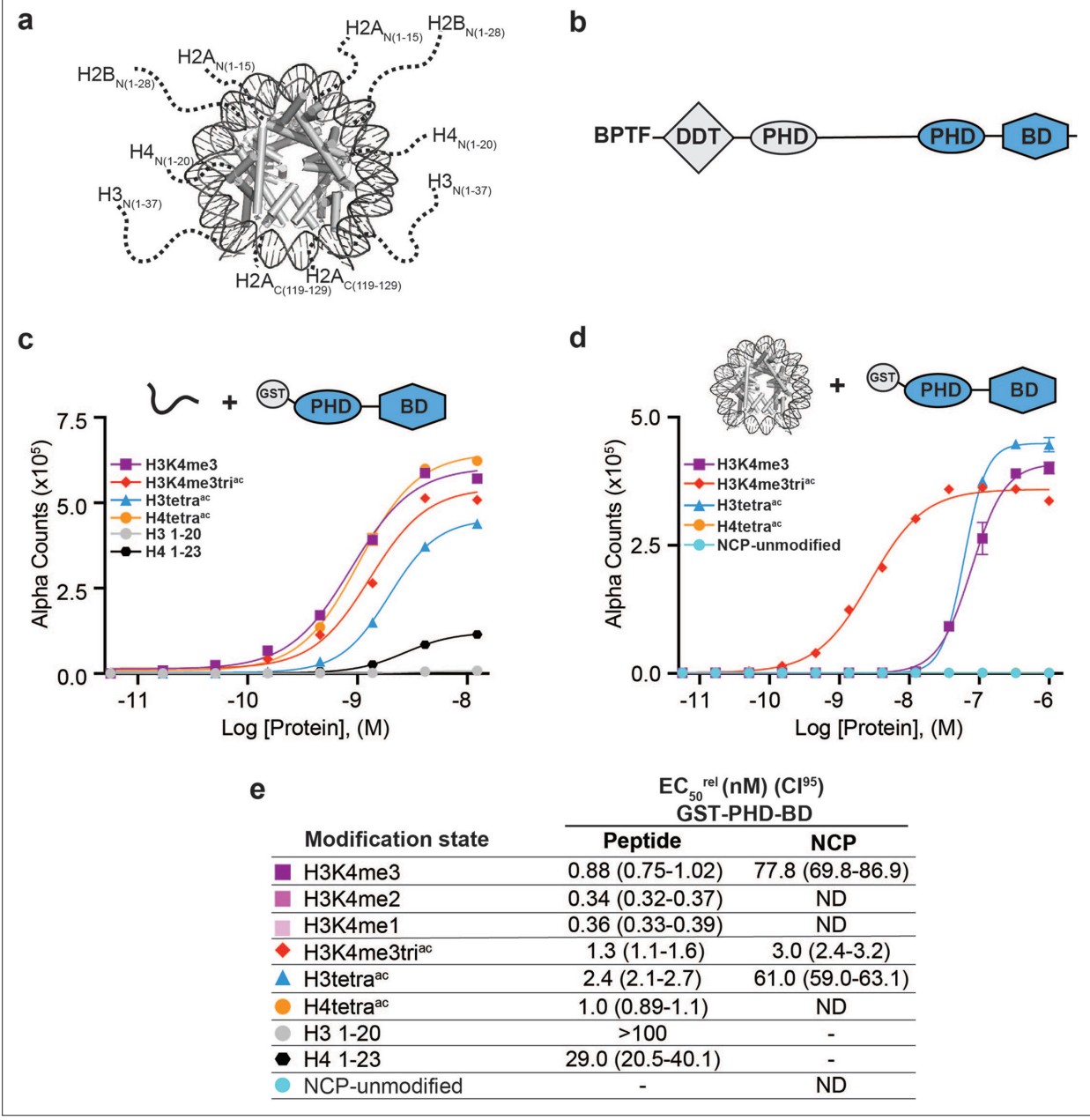

**Figure 1.** BPTF PHD-BD demonstrates restricted and synergistic PTM binding in the nucleosome vs. peptide context. (**a**) The nucleosome core particle (NCP) (PDB: 3LZ0): histone N- and C-terminal tails (as defined by trypsin digest) are depicted as dotted lines and to relative scale. (**b**) Secondary domain architecture of BPTF [Uniprot Q12830; 3,046 aa; 338 kDa]. Region covered by the C-terminal tandem PHD-BD (aa 2865–3036; as used through this study) in blue. (**c, d**) *dCypher* assay Alpha counts plotted as a function of GST-PHD-BD Query concentration to histone peptide (**c**) or NCP (**d**) Targets. (**e**) Relative EC$_{50}$ (EC$_{50}$$^{rel}$) and 95% confidence interval (CI95) values from *dCypher* curves (in **c, d**) and *Figure 1—figure supplement 1d and e*; for calculation see 'Materials and methods'. Targets are color coded as per legends. ND, Not Detected, Not Testable.

The online version of this article includes the following source data and figure supplement(s) for figure 1:

**Figure supplement 1.** Binding preference of BPTF GST-PHD-BD for peptide vs. NCP substrates.

**Figure supplement 2.** Representative protein QC.

**Figure supplement 2—source data 1.** Full native gel image for modified versaNuc reconstitution, stained with ethidium bromide.

**Figure supplement 2—source data 2.** Full native gel image for wild-type versaNuc reconstitution, stained with ethidium bromide.

**Figure supplement 2—source data 3.** Full SDS-PAGE image for modified versaNuc reconstitution, stained with coomassie blue.

**Figure supplement 2—source data 4.** Full SDS-PAGE image for wild-type versaNuc reconstitution, stained with coomassie blue.

*Figure 1 continued on next page*

*Figure 1 continued*

**Figure supplement 2—source data 5.** Full SDS-PAGE image for wild-type and mutant 6His-BD, stained with coomassie blue.

**Figure supplement 2—source data 6.** Full SDS-PAGE image for wild-type 6His-PHD-BD, stained with coomassie blue.

**Figure supplement 2—source data 7.** Full SDS-PAGE image for wild-type GST-PHD-BD, stained with coomassie blue.

**Figure supplement 2—source data 8.** Full SDS-PAGE image for wild-type 6His-PHD, stained with coomassie blue.

**Figure supplement 2—source data 9.** Full SDS-PAGE image for mutant GST-PHD and GST-PHD-BD, stained with coomassie blue.

**Figure supplement 2—source data 10.** All gel images for *Figure 1—figure supplement 2* labeled and with regions cropped for figure denoted with dashed boxes.

To establish this study, we set out to more comprehensively investigate if context alters the BPTF PHD-BD readout of histone PTMs. To this end, we screened GST- and 6His- tagged forms of the tandem reader (GST-PHD-BD and 6His-PHD-BD) against large panels of biotinylated PTM-defined peptides (287x) and nucleosome core particles (NCPs, wrapped by 147 bp DNA; 59x) using the *dCypher* approach (*Marunde et al., 2022a*) on the Alpha platform (*Eglen et al., 2008*; *Quinn et al., 2010*; *Figure 1—figure supplement 1a*). This no-wash bead-based proximity assay allows measurement of the relative $EC_{50}$ ($EC_{50}^{rel}$) between Queries: Targets (i.e. readers: histone PTMs) by plotting Alpha Counts (fluorescence) as a function of protein concentration (*Marunde et al., 2022a*; see *Supplementary file 1* for all $EC_{50}^{rel}$ in this study, and 'Materials and methods' for their means of calculation and distinction from an equilibrium $K_d$).

In agreement with previous studies (*Wysocka et al., 2006*; *Li et al., 2006*), the GST-PHD-BD Query showed strong selectivity for methylated H3K4 peptides over all other methyl-residues represented (me1-2-3 at H3K9, H3K27, H3K36, and H4K20: *Figure 1—figure supplement 1b*). Also in agreement with previously (*Ruthenburg et al., 2011*; *Nguyen et al., 2014*), GST-PHD-BD preferred acetylated H4 tail peptides, though we observed little difference in binding to a multiply acetylated tail vs. any singly acetylated residue on the same (*Figure 1c and e* and *Figure 1—figure supplement 1b*). We also observed comparable binding to singly or multiply acetylated H3 tail peptides, though with approximately twofold weaker $EC_{50}^{rel}$ as compared to H4 peptides (*Figure 1c and e* and *Figure 1—figure supplement 1b*). Similar results were obtained with a 6His-PHD-BD Query (see 'Materials and methods'). Finally, and again in agreement with previously (*Chen et al., 2020*), we observed no preference for a H3K4me3K9acK14acK18ac (hereafter H3K4me3tri[ac]) peptide over those containing each PTM class alone (*Figure 1c and e*). Thus, peptides provide no support for a 'histone code,' in which multivalent engagement by PHD-BD would be expected stronger than either individual domain to a combinatorially modified substrate.

We next examined the interaction of GST-PHD-BD with PTM-defined NCPs and found several striking differences. First, the overall affinity for nucleosomes was reduced relative to peptides (*Figure 1c–e*). Second, NCPs recapitulated only a subset of the interactions observed with peptides (*Figure 1c–e* and *Figure 1—figure supplement 1b and c*). Differences included a newfound selectivity for H3K4me3 over the me2 /me1 states (*Figure 1—figure supplement 1d and e*), and binding to acetylated H3 but not acetylated H4 (*Figure 1d and e* and *Figure 1—figure supplement 1b and c*). A third contrast to peptides was a dramatic increase in the affinity of GST-PHD-BD for NCPs containing the H3K4me3tri[ac] combinatorial vs. those containing each PTM class alone (26-fold over H3K4me3; 20-fold over H3K4acK9acK14acK18ac (hereafter H3tetra[ac])) (*Figure 1d and e*). This last point would support a 'histone code' where reader domains act synergistically to engage preferred PTM patterns.

To further refine the PTM patterns recognized by GST-PHD-BD in the nucleosome context we tested substrates containing individual acetyl-lysines. We observed a similar $EC_{50}^{rel}$ to H3K4me3tri[ac] for H3K4me3K14ac and H3K4me3K18ac, but fourfold weaker for H3K4me3K9ac (*Figure 1—figure supplement 1f and g*). Notably, crystal structures of BPTF BD in complex with acetylated histone peptides (*Ruthenburg et al., 2011*) indicate the bromodomain binding pocket can accommodate only one acetyl-lysine. Thus, data supports that PHD-BD preferentially reads out nucleosomal H3K4me3K14ac or H3K4me3K18ac.

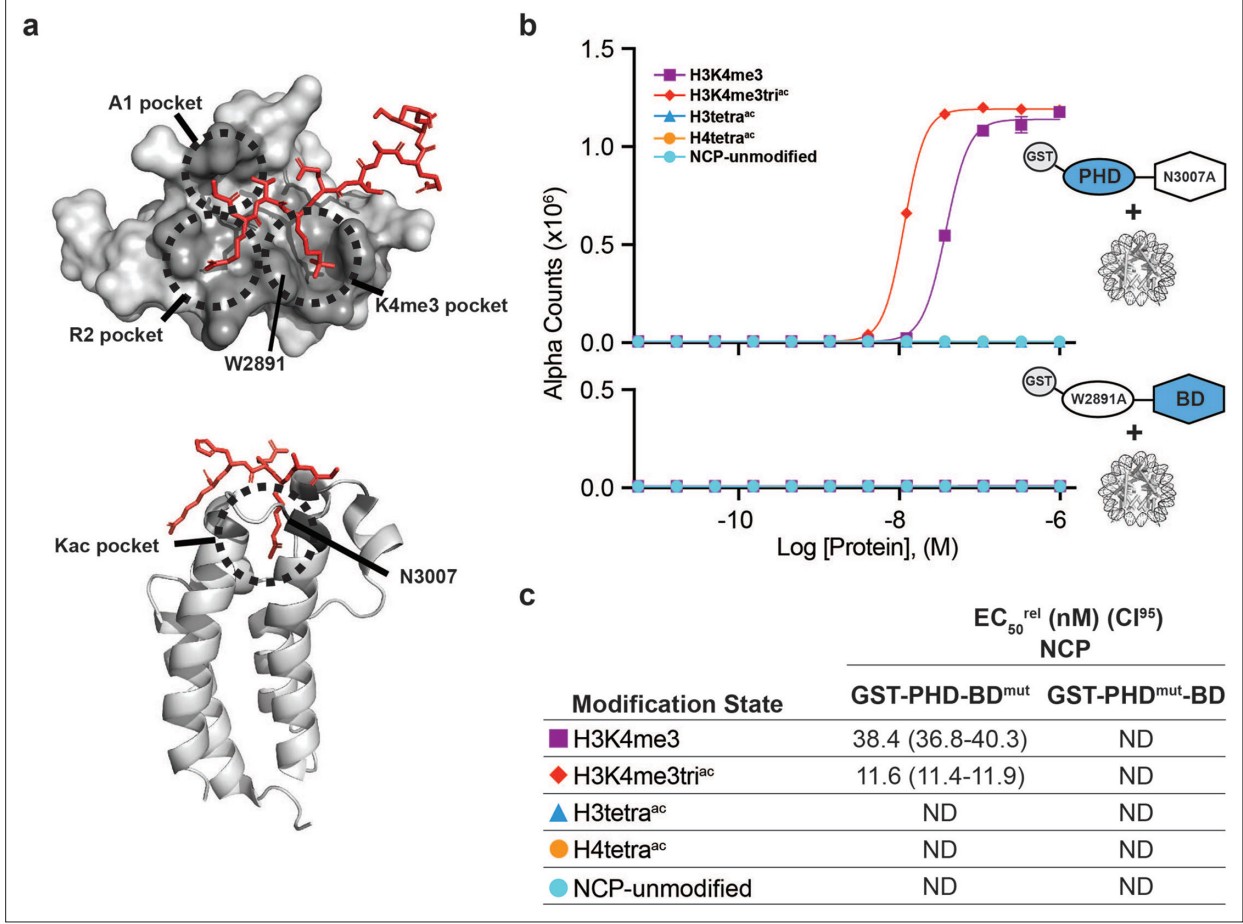

**Figure 2.** BPTF PHD and BD both contribute to nucleosome binding. (**a**) The PHD-H3K4me3 (top) and BD-Kac (bottom) binding pockets on previously solved structures of the individual domains in complex with histone peptides (PDB: 2FUU and 3QZT). Binding pockets are circled/labeled: on PHD for A1, R2, and K4me3; on BD for Kac. Relative location of PTM-binding residues W2891 (PHD) and N3007 (BD) also indicated and mutated to alanine in (**b, c**). (**b**) dCypher assay Alpha counts plotted as a function of GST-PHD-BD[N3007A] (GST-PHD-BD[mut]; top) or GST-PHD[W2891A]-BD (GST-PHD[mut]-BD; bottom) Query concentration to NCP Targets. (**c**) $EC_{50}^{rel}$ (CI95) values from dCypher curves in (**b**). Targets color coded as per legends. ND, Not Detected.

The online version of this article includes the following figure supplement(s) for figure 2:

**Figure supplement 1.** Individual BPTF reader domains have reduced affinity and restricted specificity in the nucleosome context.

**Figure supplement 2.** NMR analysis of mutant BPTF BD.

## Individual reader domains have reduced affinity and altered specificity in the nucleosome context, with PHD-BD both required for full activity of the tandem module

To further dissect the contribution of each domain to synergistic binding by PHD-BD, we tested their individual reader ability for peptides and NCPs. As for the PHD-BD tandem, the 6His-PHD affinity for NCPs was reduced relative to peptides (*Figure 2—figure supplement 1a–d* and *Supplementary file 1*). Interestingly, while 6His-PHD was preferentially associated with H3K4me3 and approximately twofold weaker to H3K4me3tri[ac] peptides, this order was inverted for NCPs (compare *Figure 2— figure supplement 1a and c* and *Figure 2—figure supplement 1b and d*). The same affinity trends were observed for GST-PHD, which favored NCPs with H3K4me3 and co-incident acetyl-lysine, but had no preference between K9ac, K14ac, or K18ac (see 'Materials and methods' and *Figure 2—figure supplement 1b and e*). Regarding the bromodomain, 6His-BD bound both acetylated H3 and H4 peptides, but with a preference for acetylated H4 (*Ruthenburg et al., 2011*; *Figure 2—figure supplement 1a and f*). However, when presented with NCPs, it failed to bind any tested targets (*Figure 2— figure supplement 1b and g*).

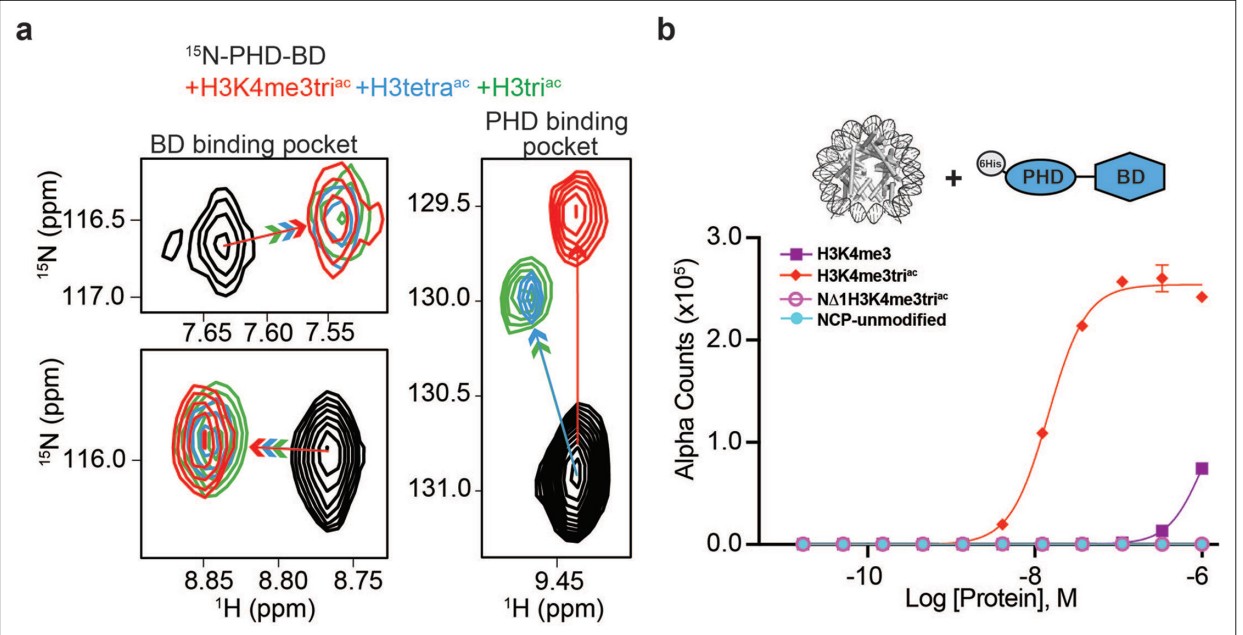

**Figure 3.** BPTF PHD-BD bind multivalently to the H3 tail. (**a**)[1]H,[15]N-HSQC overlays of [15]N-PHD-BD apo (black) or in the presence of H3tri[ac] (green), H3tetra[ac] (blue), or H3K4me3tri[ac] (red) peptides. Arrows denote trajectory of chemical shift perturbation (CSP) and are colored by peptide. Shown are representative resonances for the bromodomain (BD) (left) and plant homeodomain (PHD) (right) binding pockets. (**b**) Histone H3-A1 is essential for 6His-PHD-BD binding to nucleosome core particles (NCPs) (compare H3K4me3tri[ac] to NΔ1H3K4me3tri[ac] [integrity of each target confirmed with anti-H3K4me3 (*Figure 3—figure supplement 1d*)]). *dCypher* assay Alpha counts are plotted as a function of Query concentration to indicated NCP Targets.

The online version of this article includes the following figure supplement(s) for figure 3:

**Figure supplement 1.** BPTF PHD-BD multivalent association with the H3 tail.

To further investigate the contribution of each domain to tandem activity, we created individual point mutants of the PHD (aromatic cage W2891A; PHD[mut]) or BD (ZA-loop N3007A; BD[mut]) (*Figure 2a* and *Supplementary file 2A*) to remove functionality but retain domain structure (*Wysocka et al., 2006*; *Li et al., 2006*; *Figure 2—figure supplement 2*). On NCPs, GST-PHD[mut]-BD lost binding to all tested targets (including H3K4me3, H3tetra[ac], or H3K4me3tri[ac]), while GST-PHD-BD[mut] bound H3K4me3 weaker than H3K4me3tri[ac] but did not associate with H3tetra[ac] (*Figure 2b and c*). This revealed that even in the tandem context a functional BD is insufficient to mediate NCP binding without a functional PHD.

Thus nucleosome context impacts the BPTF PHD-BD interaction with modified histone tails in a manner that would not be predicted by individual reader domain or histone peptide studies. PHD alone bound H3K4me3, but preferred this in context of H3 tail acetylation and without distinguishing individual acetylated residues (K9ac, K14ac, or K18ac; *Figure 2—figure supplement 1b and e*). BD alone failed to associate with any NCPs. but when partnered with its endogenous PHD (wild-type) engaged the H3tetra[ac] tail, but not individual acetylated forms (*Figure 2b* and *Figure 1—figure supplement 1c, f and g*). Finally, PHD-BD showed a >20-fold preference for H3K4me3tri[ac] over H3K4me3 or H3tetra[ac] (*Figure 1d and e*) and fourfold preference for H3K4me3 paired with H3K14ac or H3K18ac over H3K9ac (*Figure 1—figure supplement 1f and g*).

### The PHD-BD makes multivalent contacts with the acetylated H3 tail

As above, in the tandem context, the PHD supports BD association with the H3tetra[ac] tail even where H3K4 is unmethylated. To investigate this, we used NMR spectroscopy to record sequential [1]H,[15]N-HSQC spectra on [15]N-labeled PHD-BD after addition of unlabeled H3tri[ac], H3tetra[ac], or H3K4me3tri[ac] peptides (*Figure 3a* and *Figure 3—figure supplement 1a–c*). Chemical shift perturbations (CSPs) in BD resonances were observed on addition of all three peptides, indicating ligand engagement. Further, the bound state chemical shift was similar for all three peptides, suggesting an association mechanism independent of H3K4 modification state (*Figure 3a*). However, for PHD resonances the

H3K4 modification state elicited distinct CSPs, with H3tetra$^{ac}$ and H3tri$^{ac}$ showing nearly identical bound state chemical shifts vs. that for H3K4me3tri$^{ac}$ (*Figure 3a*). Together this reveals that PHD-BD associates with the acetylated H3 tail likely in a multivalent manner, employing both domains independent of H3K4 modification status, but forming a unique complex when H3K4 is trimethylated.

The PHD: H3 binding interface includes pockets for histone residues A1, R2, and K4me3 (*Li et al., 2006*; *Figure 2a*), with the last needed for robust NCP interaction by an isolated PHD. From our NMR data, we hypothesized the A1 and/or R2 interactions contribute to PHD-BD association with the acetylated H3 tail. To test this, we truncated A1 in the context of H3K4me3tri$^{ac}$ and observed PHD-BD was unable to bind the resulting NCP (NΔ1) (*Figure 3b*). Thus, recognition of the H3 N-terminus is critical for BPTF PHD engagement, an observation consistent with the binding mechanism for other PHD fingers (*Musselman et al., 2009*).

## In the nucleosome context, DNA interactions occlude the H4 tail and alter reader engagement

We next asked how the BPTF BD interaction with acetylated H4 might be abrogated in nucleosomes, despite robust binding to comparable peptides (*Figure 1*, *Figure 1—figure supplement 1* and *Figure 2—figure supplement 1*). Reduced reader binding to NCPs relative to peptides in *dCypher* (e.g. *Figure 2—figure supplement 1* and *Supplementary file 1*) is consistent with our NMR studies showing strong inhibition of PHD binding to H3K4me3 in the nucleosome context (*Morrison et al., 2018*). There we demonstrated that H3 tail occlusion is due primarily to K/R interactions with the nucleosomal DNA backbone (*Morrison et al., 2018*; *Ghoneim et al., 2021*). We thus explored if a similar mechanism operated for the H4 tail.

The H4 tail is K/R-rich, has decreased dynamics in the nucleosome vs. peptide context, and computational models suggest it may also form a fuzzy complex with DNA (*Rabdano et al., 2021*). To explore this further, we used NMR spectroscopy with an NCP containing $^{15}$N-H4 (*Figure 4—figure supplement 1*). Due to its large size (~200 kDa) and resultant slow tumbling, only very flexible regions (such as the tails) should be NMR observable using this isotope labeling scheme (*Figure 1a*). Consistent with previously (*Rabdano et al., 2021*; *Zhou et al., 2012*; *Kim et al., 2023*) we observed resonances for only 15 of the 101 non-proline amino-acids of H4, corresponding to tail residues 1–15 (*Figure 4*). However, this represents only 15/20 possible resonances (assuming fast exchange on the NMR time-scale) for the H4 N-terminal tail (as classified by trypsin accessibility; e.g. *Figure 1a*; *Böhm and Crane-Robinson, 1984*). The severe line-broadening observed for residues 16–20 (the H4 tail basic patch: *Figure 4—figure supplement 2*) indicates this region is likely stably associated with the nucleosome core, in agreement with previous structural and biochemical studies (*Zheng and Hayes, 2003*) However, the conformation of H4 residues 1–15 is less clear.

To further investigate any conformational differences between a free tail and that in the nucleosome context, we generated $^{15}$N-H4 (1-25) in peptide form. Overlay of the resulting NMR spectra showed CSPs in every H4 tail resonance when comparing peptide and NCP (*Figure 4a*), consistent with a context-dependent conformation. We next collected sequential $^{1}$H,$^{15}$N-HSQC spectra of the $^{15}$N-H4 (1-25) peptide upon addition of unlabeled DNA (*Figure 4b*), and observed CSPs for every resonance, confirming the H4 tail bound DNA, and every residue is impacted. Overlay of the DNA-bound $^{15}$N-H4 peptide and $^{15}$N-H4-NCP spectra showed very similar chemical shifts, consistent with the entire H4 tail associating with nucleosomal DNA (*Figure 4c*), and in-line with previous cross-linking and molecular dynamics simulation studies (*Murphy et al., 2017*; *Karch et al., 2018*; *Mullahoo et al., 2020*). The differential linewidth of resonances indicates the H4 tail has two distinct dynamic regions: residues 1–15 likely exchange quickly between multiple conformations on DNA, consistent with a fuzzy complex *Fuxreiter, 2018*; *Tompa and Fuxreiter, 2008*; while residues 16–20 (the basic patch) exchange much more slowly and/or between fewer states, leading to signal loss. This is distinct from the H3 tail, where every residue experiences fast dynamics consistent with a fuzzy complex. The different behavior of the tails may be related to charge distribution and/or positioning relative to the NCP core.

The above data supports that, similar to the H3 tail, the H4 tail conformation in the nucleosome context occludes accessibility, and potentially explains the loss of BPTF BD/PHD-BD association with acetylated H4 in *dCypher* (*Figure 1c–e*). To investigate this further, we generated a $^{15}$N-H4K$_{5}$16ac-NCP. Relative to unmodified NCP, the acetylated NCP spectra had additional peaks (*Figure 4d*,

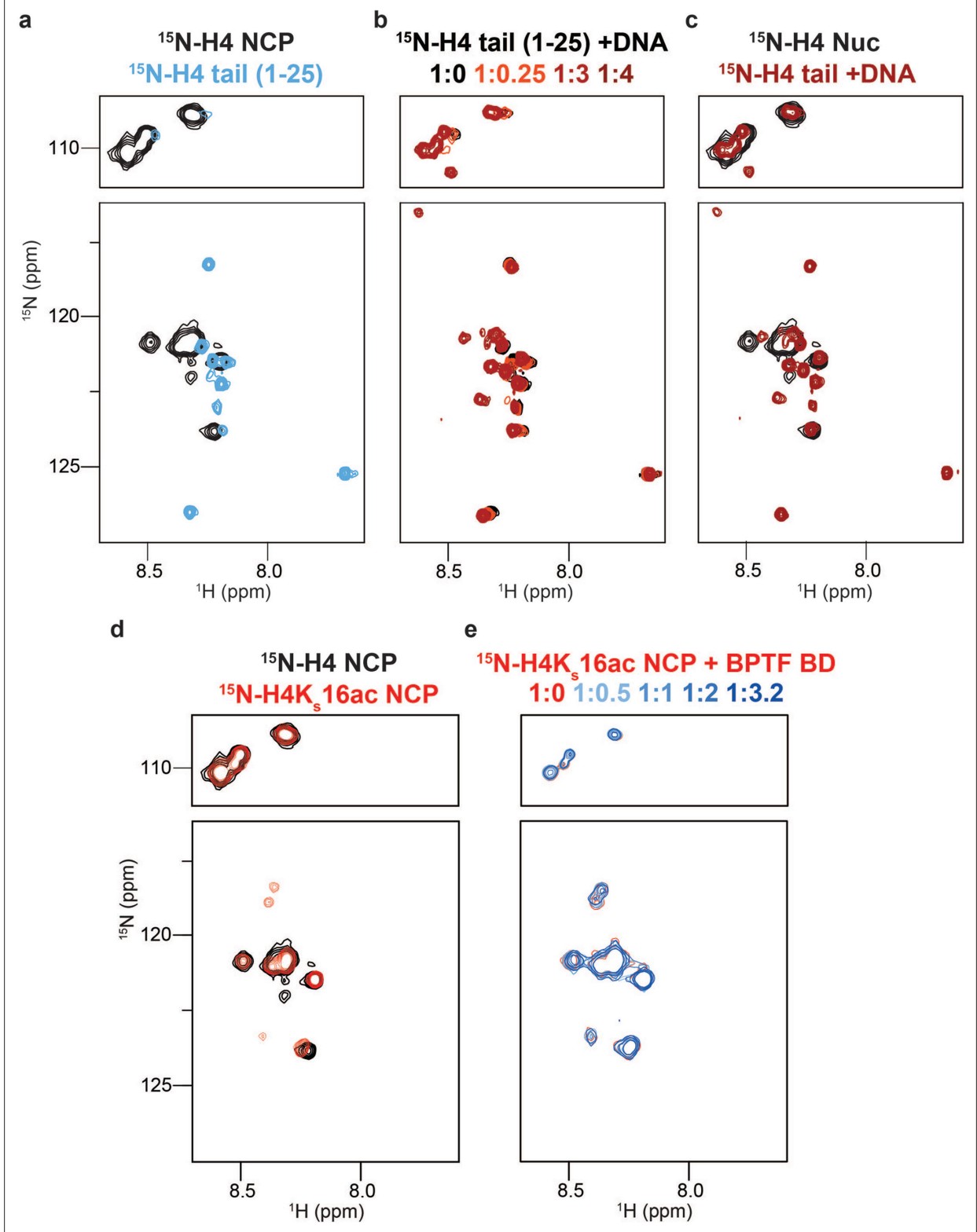

**Figure 4.** DNA binding occludes BD access to the H4 tail in the nucleosome context. (**a**) Overlay $^1H,^{15}N$-HSQC spectra of $^{15}N$-H4-NCP (black) and $^{15}N$-H4-tail peptide (residues 1–25, blue). (**b**) Overlay $^1H,^{15}N$-HSQC spectra of $^{15}N$-H4-tail peptide upon titration of a 21 bp double-stranded DNA. Molar ratios are denoted by color in legend. (**c**) Overlay $^1H,^{15}N$-HSQC spectra of $^{15}N$-H4-NCP (black) and $^{15}N$-H4-tail peptide saturated with DNA (red). (**d**) Overlay $^1H,^{15}N$-HSQC spectra of $^{15}N$-H4-NCP (black) and $^{15}N$-H4$K_s$16ac-NCP (red). (**e**) Overlay $^1H,^{15}N$-HSQC spectra of $^{15}N$-H4$K_s$16ac-NCP (red) upon

*Figure 4 continued on next page*

*Figure 4 continued*

titration of BPTF BD (blue). Molar ratios are denoted by color in legend. Note that contour levels are adjusted in (**e**) relative to (**d**) for visualization purposes.

The online version of this article includes the following source data and figure supplement(s) for figure 4:

**Figure supplement 1.** Representative protein QC (NMR related).

**Figure supplement 1—source data 1.** Full native gel image for $^{15}$N-H4-NCP reconstitution, stained with ethidium bromide.

**Figure supplement 1—source data 2.** Full SDS-PAGE image for $^{15}$N-H4-NCP reconstitution, stained with coomassie blue.

**Figure supplement 1—source data 3.** All gel images for *Figure 4—figure supplement 1* labeled and with regions cropped for figure denoted with dashed boxes.

**Figure supplement 2.** Nucleosome crystal structure (PDB ID: 3LZ0) showing interaction of the H4 tail basic patch (residues: K16-R17-H18-R19-K20) with DNA.

**Figure supplement 3.** Representative QC (H4K$_s$16ac related).

**Figure supplement 3—source data 1.** Full SDS-PAGE image of histone H4K16C protein purification for H4K$_s$16ac installation.

**Figure supplement 3—source data 2.** Full SDS-PAGE image for $^{15}$N-H4K$_s$16ac-NCP reconstitution, stained with coomassie blue.

**Figure supplement 3—source data 3.** Full native gel image for $^{15}$N-H4K$_s$16ac-NCP reconstitution, stained with ethidium bromide.

**Figure supplement 3—source data 4.** Full SDS-PAGE image for BPTF BD, stained with coomassie blue.

**Figure supplement 4.** BRD4 GST-BD1 binds acetylated histone H4 tail peptides and NCPs.

*Figure 4—figure supplement 3*), suggesting this modification weakens the H4 tail interaction with DNA (similar to previously observed with the charge neutralization mimetic H4K16Q *Zhou et al., 2012*). However, the broadness of the peaks suggests the acetylated H4 basic region (16-20) still interacts more robustly with DNA than H4 tail residues 1–15 (or indeed the H3 tail). Titration of BPTF BD into the $^{15}$N-H4K$_s$16ac-NCP did not lead to any significant CSPs, supporting that H4K16ac was occluded from binding by this reader in the NCP context (*Figure 4e*).

To investigate if the nucleosome abrogates all interactions with the H4 tail we turned to an alternate bromodomain Query (GST-BRD4-BD1; *Supplementary file 2A*). BRD4-BD1 has previously been shown to bind acetylated H4 tail peptides *Filippakopoulos et al., 2012*, and *dCypher* confirmed the strongest EC$_{50}^{rel}$ for H4tetra$^{ac}$ over all peptides tested (*Figure 4—figure supplement 3a and c*). In contrast to BPTF BD, BRD4-BD1 also bound H4tetra$^{ac}$ in the nucleosome context, though with weaker affinity than the comparable peptide (EC$_{50}^{rel}$ 7.4 nM NCP vs. 0.7 nM peptide; *Figure 4—figure supplement 4b and c*). Thus, nucleosomal H4 tail accessibility is reader-dependent (also recently demonstrated for PHIP BD1-BD2 *Morgan et al., 2021*), and the ability to bind may rely on several factors including overall affinity or different engagement mechanisms. For instance, BRD4 BD1 (unlike BPTF BD) can associate with DNA (*Miller et al., 2016*; *Kalra et al., 2022*), and such competition may help disengage the H4 tail from the nucleosome core.

Together, this suggests that to enable binding in the nucleosome context a reader must be able to displace the modified histone tail from DNA. Tail accessibility can be enhanced by disrupting the DNA interaction via modification of sidechain charge (*Stützer et al., 2016*; *Morrison et al., 2018*), as where distal acetylation of the H3 tail improved BPTF PHD engagement with H3K4me3 (*Figure 2—figure supplement 1b, d and e*). Notably, acetylation does not fully release the tail from DNA binding (*Morrison et al., 2018*; *Mutskov et al., 1998*), such that the PHD still showed weaker association with the H3K3me3tri$^{ac}$ NCP relative to peptide. This may also explain why BPTF BD alone was insufficient to engage the acetylated H3 or H4 tails in the nucleosome context, since as a weaker binder it cannot effectively displace even acetylated tails to engage its reader function.

## BPTF PHD-BD interacts with nucleosomal H3K4me3tri$^{ac}$ in <u>cis</u> > trans

All nucleosome data above were with homotypic NCPs, where both H3 proteins in the octamer were identically modified. As such they cannot address the relative contribution of BPTF PHD-BD binding to their target PTMs in cis or trans. To definitively explore this we synthesized fully defined heterotypic NCPs ('Materials and methods' and *Jain et al., 2023*) where PTMs could be independently distributed across each H3 tail (e.g. [H3K4me3K14ac • H3] vs. [H3K4me3 • H3K14ac]). In *dCypher* assays GST-PHD-BD showed 24-fold stronger binding to the *cis* vs. *trans* combinatorial context (EC$_{50}^{rel}$ 10.8 nM [H3K4me3K14ac • H3] vs. 263 nM [H3K4me3 • H3K14ac]; *Figure 5a and b*). Furthermore, binding

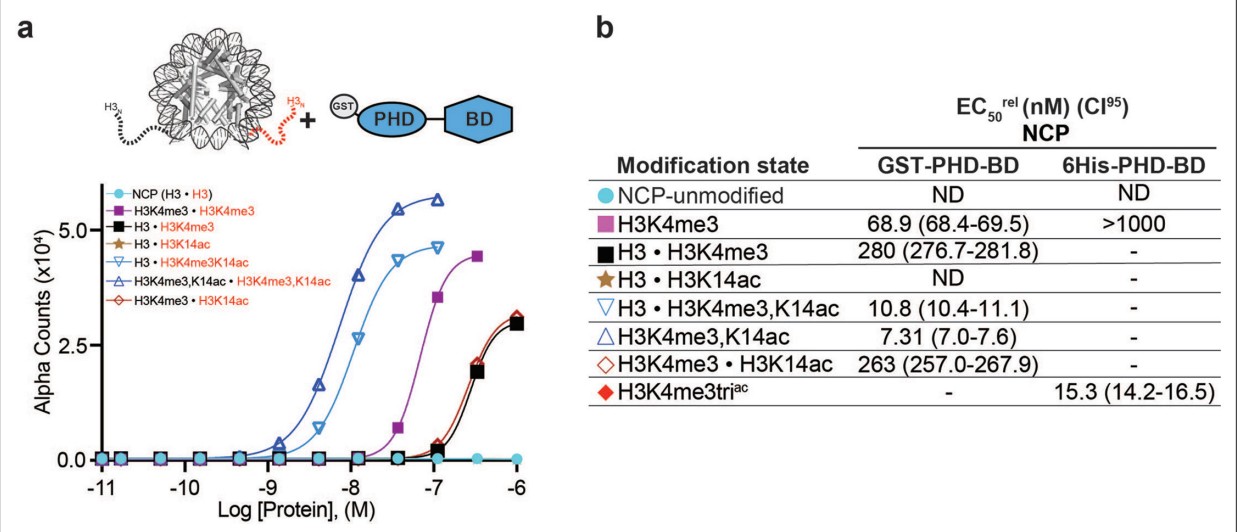

**Figure 5.** BPTF PHD-BD binds its target PTMs on the H3 tail in cis vs. trans. (**a**) *dCypher* assay Alpha counts plotted as a function of GST-PHD-BD Query concentration to homotypic (e.g. [H3 • H3]) or heterotypic (e.g. [H3 • H3K4me3]) NCP Targets (created as in 'Materials and methods'). (**b**) Relative $EC_{50}$ ($EC_{50}^{rel}$) and 95% confidence interval (CI95) values from *dCypher* curves (in **a**; for calculation see 'Materials and methods'). • indicates a heterotypic Target. Limited testing data from a 6His-PHD-BD query added for comparison. Targets are color coded as per legends. ND, Not detected; -, Not tested.

to the *trans* combinatorial NCP was indistinguishable from [H3K4me3 • H3], suggesting the tandem reader requires both PTMs in cis, and trans tail engagement is of minimal contribution. Supporting this interaction mechanism, GST-PHD-BD had only slightly improved binding to homotypic H3K4me3K14ac over heterotypic [H3K4me3K14ac • H3] ($EC_{50}^{rel}$ 7.3 nM vs. 10.8 nM; *Figure 5a and b*), which would be expected if reader engagement to each nucleosomal H3 tail is an essentially independent event.

## BPTF PHD-BD promotes a specific association with H3K4me3tri[ac] in chromatin

The above data demonstrates that BPTF PHD-BD preferentially associates with nucleosomal H3K4me3K14ac or H3K4me3K18ac in vitro. To investigate if this preference is recapitulated on chromatin, we performed CUT&RUN with antibodies to BPTF, H3K4me3, and H3K18ac in K562 cells ('Materials and methods' and *Supplementary file 2*). This identified extensive genomic co-localization of BPTF with each PTM, but the greatest degree of overlap when both are present (*Figure 6—figure supplement 1*). As a bulk analysis, CUT&RUN is unable to confirm definitive co-enrichment of all elements, with one possible interpretation that these data represent distinct sub-populations. We thus designed a new approach (Reader CUT&RUN; 'Materials and methods') where GST-PHD-BD was complexed with an antibody to GST (α-GST) to create a CUT&RUN compatible reagent. We also developed DNA-barcoded PTM-defined NCPs (unmodified, H3K4me3, H3tetra[ac], and H3K4me3tri[ac]; *Figure 6a*) as a CUT&RUN spike-in to monitor assay performance and the GST-PHD-BD preference in situ. In these controlled studies GST-PHD-BD showed a dramatic preference for spike-ins containing the combinatorial signature (H3K4me3tri[ac]) relative to each PTM alone (sixfold over H3K4me3, 41-fold over H3tetra[ac]; *Figure 6b*), recapitulating our *dCypher* observations (e.g. *Figure 1d*). The genomic enrichment of GST-PHD-BD further confirmed its combinatorial preference, with binding regions showing extensive overlap with those containing H3K4me3 and H3K18ac (*Figure 6c and d*). Furthermore, the genomic enrichment of GST-PHD-BD was also highly correlated with that of endogenous BPTF (*Figure 6c and d*), supporting that the tandem reader domains are sufficient to drive effective in vivo localization.

## Discussion

Taken together, our data indicate that nucleosome context strongly influences reader domain engagement with histone PTMs. Previous studies have described reduced reader affinity towards nucleosomes

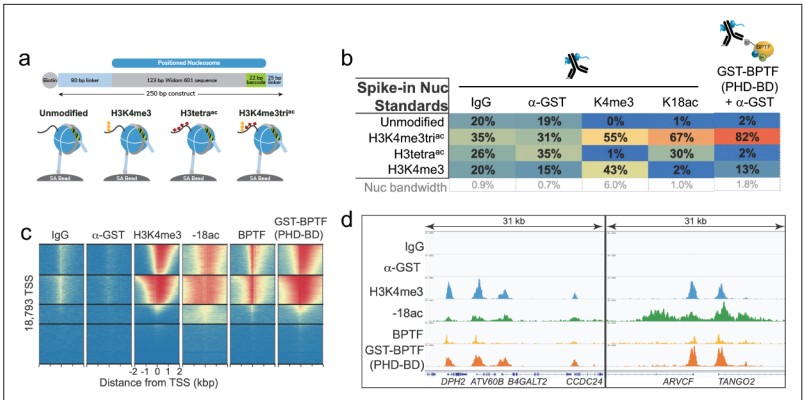

| Spike-in Nuc Standards | IgG | α-GST | K4me3 | K18ac | GST-BPTF (PHD-BD) + α-GST |
|---|---|---|---|---|---|
| Unmodified | 20% | 19% | 0% | 1% | 2% |
| H3K4me3tri$^{ac}$ | 35% | 31% | 55% | 67% | 82% |
| H3tetra$^{ac}$ | 26% | 35% | 1% | 30% | 2% |
| H3K4me3 | 20% | 15% | 43% | 2% | 13% |
| Nuc bandwidth | 0.9% | 0.7% | 6.0% | 1.0% | 1.8% |

**Figure 6.** The in vitro combinatorial preference of BPTF PHD-BD is recapitulated in vivo. (**a**) CUTANA nucleosome spike-ins contain a 5'biotin for immobilization to magnetic beads and a DNA barcode to define post-translational modification (PTM) status/monitor release into the CUT&RUN eluate. A four-member panel was assembled to explore GST-PHD-BD binding (unmodified, H3K4me3, H3tetra$^{ac}$, H3K4me3tri$^{ac}$; on 80-<u>N</u>-25 DNA containing a central 147 bp 601 <u>N</u>ucleosome positioning sequence with embedded 22 bp DNA barcode). (**b**) GST-PHD-BD shows strong preference for spike-in nucleosome containing H3K4me3tri$^{ac}$. Table shows relative release of spike-ins (percent barcoded nucleosome/total barcode reads) in Reader CUT&RUN ('Materials and methods'). Antibodies are noted by column; GST-BPTF (PHD-BD) is detected by α-GST to facilitate pAG-MNase recruitment. 'Nucleosome bandwidth' is the percentage of total sequence reads taken up by spike-in standards. (**c**) Heatmap of CUT&RUN signal aligned to the transcription start site (TSS, +/−2 kb) of 18,793 genes in K562 cells. Rows were k-means clustered into four groups (boxed) using ChAsE chromatin analysis tool (***Younesy et al., 2016***). High and low signal (red and blue, respectively) are ranked by / linked to H3K4me3 (top to bottom). (**d**) CUT&RUN RPKM normalized tracks at representative loci using Integrative Genomics Viewer (IGV, Broad Institute). Note the co-localization of BTPF (endogenous) or GST-PHD-BD (exogenous) with H3K4me3 and H3K18ac; that H3K18ac alone is insufficient to recruit BTPF or GST-PHD-BD; and that GST-PHD-BD shows robust recruitment at some locations where BPTF is absent (e.g. *B4GALT2* promoter; see 'Discussion').

The online version of this article includes the following figure supplement(s) for figure 6:

**Figure supplement 1.** The combinatorial PTM preference of BPTF PHD-BD in vitro is mirrored by its in vivo co-localization.

relative to histone tail peptides (***Morrison et al., 2018***; ***Wang and Hayes, 2007***; ***Gatchalian et al., 2017***; ***Peng et al., 2021***), but here we show that the engaged PTM(s) may also be restricted (e.g. loss of BPTF BD and PHD-BD binding to H4tetra$^{ac}$; *Figure 1c–e* and *Figure 2—figure supplement 1a–b, f–g*), or the preferred PTM pattern may be altered (e.g. BPTF PHD having stronger binding to nucleosomal H3K4me3 with coincident tail acetylation; *Figure 2—figure supplement 1b and e*). We propose this is due (at least in part) to the default association of nucleosomal histone tails with DNA (***Ghoneim et al., 2021***), which limits accessibility and requires reader domains to compete for tail engagement. As a result, histone PTMs may play multiple roles; weakening the DNA association to increase access for reader domains, providing a platform for reader domain binding, or both. We note this interaction model is largely based on in vitro studies with single nucleosomes (***Stützer et al., 2016***; ***Morrison et al., 2018***; ***Rabdano et al., 2021***; ***Zhou et al., 2012***; ***Musselman and Kutateladze, 2022***; ***Ohtomo et al., 2021***; ***Ohtomo et al., 2023***; ***Zandian et al., 2021***; ***Furukawa et al., 2022***; ***Jennings et al., 2023***; ***Furukawa et al., 2020***; ***Morrison et al., 2021***), and thus does not fully capture the chromatin environment. However, via solid-state NMR spectroscopy, a nearly identical conformation of histone tails has also been observed in chromatin arrays (***Musselman and Kutateladze, 2022***; ***Shi et al., 2018***; ***Shi et al., 2020***; ***Gao et al., 2013***; ***Xiang et al., 2018***), and the H3 tail: DNA interaction has been observed in vivo by ChIP-exo (***Rhee et al., 2014***). As such, although the full nuclear context is definitively more complex, tail displacement is almost certain to be one factor.

Given occluded tail conformation in the nucleosome context, multivalent engagement of tandem domains is unlikely to be simply defined by raw potential (i.e. the sum of individually preferred PTMs), but also by binding opportunity. For BPTF PHD-BD, this manifests as a nucleosomal restriction on H4ac tail binding, and a selectivity for H3ac. Note that here we did not test a H3K4me3, H4K16ac combinatorial nucleosome to explore the previously reported BPTF PHD-BD preference (***Ruthenburg***

*et al., 2011*). However, we do not intend to indicate that other combinations are not possible, but rather highlight a combination that was not predicted by peptide testing. We also note that H3K4me3 is invariably seen in (and most effectively created in) the *cis* context of H3 tail acetylation (see below) (*Jain et al., 2023*). As such the combinatorial H3 binding target we have identified for BPTF PHD-BD is also a preferred PTM pattern in vivo, and any nucleosome that contains H3K4me3 and H4K16ac might also be expected to contain H3 tail acetylation.

We observe multiple ways to combine multivalent contacts along the H3 tail, and thus support productive engagement. In the case of the BPTF PHD-BD tandem, the PHD can associate with H3 A1, R2, and K4me3 (*Figure 2a*, *Figure 3a and b*), while the BD can bind K14ac or K18ac (*Figure 1—figure supplement 1f–g*). Notably, when the H3 tail is only acetylated (as in the H3tetra$^{ac}$ NCP) the resulting weakening of the tail/DNA interaction combined with BD binding to Kac and PHD finger binding to A1 and R2 together support weak engagement. Alternatively, for H3K4me3 absent any acetylation, PHD contacts with A1, R2, and K4me3 also support weak NCP engagement. Finally, strong binding occurs when H3K4me3 and H3K14ac or H3K18ac are present, promoting tail displacement and allowing both the PHD and BD to most effectively engage. Thus, within a tail displacement model, tandem domains can accommodate multiple distinct PTM signatures to engage modified nucleosomes. Notably, and as seen here, these may have varying strengths of interaction which in turn may mediate an array of responses within the chromatin landscape, including differences in CAP retention at particular sites, or stabilization at an intermediate modified state. The preference of BPTF PHD-BD for H3K4me3 with H3K14ac or H3K18ac over H3K9ac may be due to the in cis proximity of K9 to H3K4me3, restricting BD binding when the PHD finger is engaged. A corollary may also be important in the preferred cis vs. trans engagement with histone tails (*Figure 5*).

Our observations on co-incident tail acetylation promoting H3K4 accessibility to BPTF PHD reader (this study) are also exhibited by MLL1C methyltransferase (*Jain et al., 2023*). There we identified a H3 acetylation-mediated switch that increases tail accessibility in the nucleosome context, and thus improves MLL1C-mediated H3K4 methylation. Evidence for this model includes an increased $K_{cat}$ (enhanced tail accessibility leads to an increase in substrate concentration), and that H3 acetylation enhances MLL1C activity in cis but has no impact in trans (demonstrated with heterotypic nucleosomes). Furthermore, middle-down MS on bulk chromatin shows that H3 tail acetylation invariably accompanies cis H3K4 methylation. Together these studies strongly support that tail occlusion can have a strong regulatory effect on the epigenome.

When moving from the peptide to nucleosome context we (and others) consistently observe individual reader domains to show reduced affinity and restricted specificity (*Jain et al., 2023*; *Morrison et al., 2018*; *Wang and Hayes, 2007*; *Gatchalian et al., 2017*; *Peng et al., 2021*; *Marunde et al., 2022a*; *Morgan et al., 2021*). An exception to this is readers with intrinsic DNA binding ability, such as the PWWPs. These form multivalent interactions with DNA and histone tails (so peptide studies are often uninformative) (*Eidahl et al., 2013*; *van Nuland et al., 2013*; *Wang et al., 2020*; *Dukatz et al., 2019*; *Tian et al., 2019*; *Weinberg et al., 2019*; *Dilworth et al., 2022*), but may also act to directly compete for the DNA, thus promoting tail accessibility when the target engaged (*Weaver et al., 2018*). Indeed, several mechanisms for modulating histone tail conformation can be imagined (*Ghoneim et al., 2021*). Beyond in cis modification of the target histone tail (as in this study), modification of an adjacent tail may alter the dynamics of the target, such trans-tail crosstalk being recently reported for H3 and H4 (*Furukawa et al., 2020*). Adjacent DNA binding domains within the same protein or complex may also play a role in displacing the target tail from DNA. Alternatively, histone tail accessibility can be modulated by changes to the canonical nucleosome composition, such as hexasomes depleted of one H2A-H2B dimer (*Morrison et al., 2021*).

In reader-CUT&RUN GST-PHD-BD recapitulated the *dCypher* preference for spike-in Nucs containing the combinatorial target (H3K4me3tri$^{ac}$) over each PTM class alone (*Figure 6b*). Furthermore, GST-PHD-BD localization across the genome was highly correlated with regions that also contain H3K4me3, H3K18ac, and endogenous BPTF (*Figure 6c–d*). Together, this suggests that the combinatorial readout of these PTMs is indeed a discerning factor in the genomic localization of BPTF: the activity of both domains is clearly important to achieve robust interaction, and thus at minimum critical to achieve proper kinetics on chromatin.

In an extended analysis of our genomics data, we considered that full-length BPTF (endogenous) could harbor additional regulatory potential over BPTF PHD-BD (exogenous). In this regard, while the

dominant signature was where H3K4me3/H3K18ac co-localized with both endogenous and exogenous (*Figure 6c–d*), we observed numerous locations where the PTM combinatorial overlapped only with exogenous (as at *B4GALT2* in *Figure 6d*), while the contrasting pattern (PTMs only overlapped with endogenous) was a much rarer species. This may be due to the relative level of exogenous to endogenous protein (or the target PTMs), where one might expect a higher abundance of exogenous to extend to locations of lower PTM density. However, peak structure comparison does not appear to support this explanation, as sites retaining exogenous but lacking endogenous are not the weakest $H3K4me3^+/H3K18ac^+$ locations. We speculate a more interesting possibility: endogenous BPTF is subject to regulation that further refines its chromatin localization beyond the simple availability of H3K4me3/H3K18ac for its C-terminal PHD-BD. Indeed, there are increasing examples of auto-regulatory elements within CAPs that modulate their activity (*Guo et al., 2015*; *Ruan et al., 2015*; *Lu et al., 2015*; *Misaki et al., 2016*; *Andrews et al., 2016b*; *Harrison et al., 2016*; *Ludwigsen et al., 2017*; *Isaac et al., 2017*; *Slaughter et al., 2018*; *Zhang et al., 2019*; *Tencer et al., 2020*; *Ren et al., 2020*; *Ren et al., 2021*; *Nodelman et al., 2021*; *Weinberg et al., 2021*), suggesting that a histone code is more than the simple availability of potentially redundant positive signals.

It is becoming increasingly clear that we should interrogate the binding of readers to histone PTMs with more physiological entities: moving away from minimal-domain queries and histone peptide targets to full-length CAPs (or higher order complexes) and nucleosomes, and thus accommodate the regulatory potential on each side. Doubtless, a more thorough mechanistic understanding will reveal novel avenues to target these interactors with therapeutic intent.

## Materials and methods
### BPTF protein constructs and preparation
Human BPTF (Uniprot Q12830) PHD finger-bromodomain (PHD-BD) and PHD finger were cloned into pGEX6p with an N-terminal Glutathione S-Transferase (GST) tag and a PreScission protease cleavage site (*Supplementary file 2A* and *Figure 1—figure supplement 2*). BPTF BD with an N-terminal 6xHistidine (6His) tag and Tobacco Etch Virus (TEV) protease cleavage site was from *Addgene* (plasmid 39111). The Q5 site-directed mutagenesis kit (*New England Biolabs* [*NEB*]) was used for domain addition/removal or single amino acid substitutions. All constructs were expressed in *E. coli* BL21 (DE3) (*Thermo Fisher Scientific* or *NEB*). Cells were grown to $OD_{600}$ ~1.0 and induced with 0.8 mM IPTG at 18 °C for ~16 hr in LB (or M9 minimal media for NMR). M9 media was supplemented with vitamin (*Centrum* Adult), 1 g/L $^{15}NH_4Cl$, and 5 g/L D-glucose. For constructs containing the BPTF PHD finger all growth media and buffers were supplemented with 100 µM $ZnCl_2$. For purification of BPTF recombinants cells were lysed by sonication, and lysates were incubated with either glutathione agarose (*Thermo Fisher Scientific*) or Ni-NTA resin (*Thermo Fisher Scientific*) to respectively enrich for GST- and 6His-tagged proteins. Fusion proteins were eluted with reduced L-glutathione or imidazole as appropriate. For NMR, samples were cleaved from the GST tag using PreScission protease. All BPTF proteins were then further purified using anion exchange (Source 15Q, *GE Healthcare Life Sciences*) and size exclusion chromatography (SEC; Superdex 75, *GE Healthcare Life Sciences*). Protein concentrations were determined by UV-Vis spectroscopy.

### Histone preparation and nucleosome core particle reconstitution for NMR
Unmodified human histones H2A, H2B, and H3 (*Supplementary file 2A*) were expressed in *E. coli* Rosetta 2 (DE3) *pLysS* or BL21 (DE3) in LB media. Cells were grown to $OD_{600}$~0.4 and induced with 0.4 mM IPTG at 37 °C for either 3 hr (for H3) or 4 hr (for H2A and H2B). $^{15}N$-labeled histone H4 (*Supplementary file 2A*) was expressed in Rosetta 2 (DE3) *pLysS* cells from a pET3a vector in M9 minimal media supplemented with vitamin, 1 g/L $^{15}NH_4Cl$, and 5 g/L D-glucose. Cells were induced at $OD_{600}$~0.4 with 0.2 mM IPTG at 37 °C for 3 hr. Histones were purified from inclusion bodies as previously (*Bao et al., 2003*) and purified by ion exchange. Mass spectrometry with positive electrospray ionization (Waters Q-Tof Premier) was used to validate histones and ensure no carbamylation occurred during purification (*Figure 4—figure supplement 1*). Samples were diluted 1:2 or 1:4 in water/acetonitrile (1:1) with 0.1% (v/v) formic acid. Acquisition and deconvolution software used during data collection and analysis were MassLynx and MaxEnt respectively.

For acetyl lysine analog, $^{15}$N-labeled histone H4 with a K-to-C mutation at lysine 16 was expressed and purified similar to wild-type H4 as described above. The acetyl-mimetic residue (K$_s$16ac) was generated through radical-mediated thiol-ene addition to the thiol group of cysteine as previously described (*Li et al., 2011*; *Bryan et al., 2017*; *Dhall et al., 2016*; *Dhall et al., 2014*). Briefly, lyophilized H4K16C protein was resuspended to a concentration of 1 mM in de-gassed reaction buffer (a 200 mM acetic acid and 15 mM L-glutathione solution, pH 4.0 to which 5 mM azo radical initiator VA-044 (*Wako chemicals* #27776-21-2), and 50 mM N-vinylacetamide (*TCI* #5202-78-8) are added immediately before the reaction). The mixture was incubated at 45 °C for 2 hr in an anaerobic environment. Histones were then dialyzed against H$_2$O to remove small molecules and subsequently subjected to mass spectrometry as described above to confirm the conversion of the K16C residue to the K$_s$16ac (*Figure 4—figure supplement 3*).

Histone octamers were prepared as previously (*Bao et al., 2003*). In brief, equimolar ratios of purified histones were combined in 20 mM Tris pH 7.5, 6 M guanidine HCl, 10 mM DTT and dialyzed into 20 mM Tris pH 7.5, 2 M KCl, 1 mM EDTA, 5 mM β-mercaptoethanol (β-ME). Octamers were SEC purified over a Sephacryl S-200 column (*GE Healthcare Life Sciences*).

The 147 bp Widom 601 nucleosome positioning sequence (NPS) (*Lowary and Widom, 1998*) was amplified in *E. coli* using a 32-repeat plasmid (*Supplementary file 2A*). DNA was purified by alkaline lysis (*Bao et al., 2003*), the 147 bp 601 NPS excised with *EcoRV*, polyethylene glycol precipitated, and further purified over a source 15Q column (*GE Healthcare Life Sciences*).

Reconstitution of Nucleosome core particles (NCPs) with 147 bp Widom 601 DNA was by desalting (*Bao et al., 2003*). In brief, octamer and DNA were combined in equimolar amounts in 2 M KCl and desalted to 150 mM KCl using a linear gradient over ~48 hr. NCPs were heat-shocked at 37 °C for 30 min for optimal positioning and purified using a 10–40% sucrose gradient. NCP formation was confirmed by sucrose gradient profile and native PAGE (see *Figure 4—figure supplements 1 and 3*). NCP concentrations were determined by UV-vis spectroscopy (after diluting in 2 M KCl to disassemble NCPs) using the absorbance from 601 DNA (calculated $\varepsilon_{260}$=2,312,300.9 M$^{-1}$cm$^{-1}$).

## H4 tail peptide purification for NMR

The histone H4 tail (residues 1–25 followed by a C-terminal tyrosine for quantification) was expressed from pGEX6p as a fusion with an N-terminal GST tag followed by a PreScission protease cleavage site (*Supplementary file 2A*). This was overexpressed in *E. coli* BL21 (DE3) (*NEB*) grown in M9 minimal media supplemented with vitamin (Centrum daily multivitamin), 1 g/L $^{15}$NH$_4$Cl, and 5 g/L D-glucose. Cells were grown to OD$_{600}$ ~1.0 and induced with 0.5 mM IPTG at 37 °C for 4 hr. The $^{15}$N-GST-H4 peptide fusion was purified on glutathione agarose resin (*Thermofisher Scientific*), cleaved with PreScission protease (16 hr at 4 °C), and products resolved by SEC (Superdex 75 10/300; *GE Healthcare Life Sciences*). Peptide identity was validated by mass spectrometry with positive electrospray ionization (Waters Q-Tof Premier). Samples were diluted 1:2 or 1:4 in water/acetonitrile (1:1) with 0.1% formic acid. Acquisition and deconvolution software used during data collection and analysis were MassLynx and MaxEnt, respectively. $^{15}$N-H4 (1-25) peptide concentration was determined by UV-vis spectroscopy using the non-native C-terminal tyrosine.

## DNA preparation for NMR

Oligonucleotides (5'-CTCAATTGGTCGTAGACAGCT-3' and the complement 5'-AGCTGTCTACGA ACCAATTGAG-3') for DNA titration NMR were from *Integrated DNA Technologies* (*IDT*). These were annealed at 50 μM by heating to 94 °C followed by gradual cooling to room temperature (in 10 mM Tris-HCl pH 7.5, 50 mM NaCl, 1 mM EDTA). Duplex DNA was purified on a source 15Q column (*GE Healthcare Life Sciences*) and analyzed by 1% agarose gel. DNA was precipitated in ethanol, resuspended in ddH$_2$O, and concentration was determined by UV-vis and the predicted extinction coefficient ($\varepsilon_{260}$=333,804.5 M$^{-1}$ cm$^{-1}$).

## NMR spectroscopy

$^1$H-$^{15}$N heteronuclear single quantum coherence (HSQC) spectra were collected on 30 μM $^{15}$N-H4 tail peptide and 80.5 μM NCP samples in 20 mM MOPS pH 7.2, 150 mM KCl, 1 mM DTT, 1 mM EDTA, and 10% D$_2$O. Data was collected at 25 °C on a 800 MHz Bruker spectrometer equipped with a cryoprobe. Titration of the 21 bp dsDNA into $^{15}$N-H4 tail peptide was performed through the collection of

sequential $^1$H-$^{15}$N HSQC spectra on the $^{15}$N-H4 tail in the apo state and with increasing DNA concentrations (spectra collected at [peptide:DNA] molar ratios of 1:0, 1:0.1, 1:0.25, 1:0.5, 1:1, 1:2, and 1:3).

Sequential $^1$H-$^{15}$N HSQC spectra of 25 μM $^{15}$N-BD and $^{15}$N-BD (N3007A) were collected with increasing concentrations of H4K16ac tail peptide (*Supplementary file 2D*) in (50 mM potassium phosphate pH 7.2, 50 mM KCl, 1 mM DTT, 1 mM EDTA, 10% D$_2$O) at 25 °C on an 800MHz Bruker spectrometer equipped with a cryogenic probe. Concentration of the stock H4K16ac peptide was analyzed by Pierce Quantitative Fluorometric Peptide Assay (*Thermo Fisher Scientific*). Spectra were collected with [$^{15}$N-BD: H4K16ac peptide] at ratios 1:0, 1:0.5, 1:1, 1:2.5, 1:5, 1:10, 1:20, 1:30, 1:50, 1:70 and [$^{15}$N-BD (N3007A): H4K16ac peptide] at ratios 1:0, 1:5, 1:20, 1:40. Sequential $^1$H-$^{15}$N HSQC spectra of 50 μM $^{15}$N-PHD-BD were collected with increasing concentrations of histone tail peptides (H3K4me3tri$^{ac}$, H3tetra$^{ac}$ or H3tri$^{ac}$: ratios 1:0, 1:0.1, 1:0.5, 1:1, 1:2, 1:4, and 1:8) in (50 mM potassium phosphate pH 7.2, 50 mM KCl, 1 mM DTT, 25 μM ZnCl$_2$, 10% D$_2$O) at 25 °C on an 800 MHz Bruker spectrometer equipped with a cryogenic probe. Sequential $^1$H-$^{15}$N HSQC spectra of 86 μM $^{15}$N-H4K$_s$16ac NCP were collected with increasing concentrations of BPTF BD. Spectra were collected with [$^{15}$N-H4Ks16ac NCP: BD] at ratios of 1:0, 1:0.5, 1:1, 1:2, and 1:3.2 in 20 mM MOPs pH 7.0, 1 mM DTT, 1 mM EDTA, 150 mM KCl, 10% D$_2$O at 25 °C on an 800 MHz Bruker spectrometer equipped with a cryogenic probe. All NMR data was processed using NMRPipe (*Delaglio et al., 1995*) and analyzed using CcpNmr Analysis (*Vranken et al., 2005*).

## Histone peptides for *dCypher*

All histone peptides for *dCypher* (*Supplementary file 2B*) were synthesized with a terminal Biotin (location as indicated) and identity was confirmed by mass spectrometry.

## Semi-synthetic nucleosomes with defined (PTMs)

PTM-defined histones, octamer,s and nucleosomes [dNucs or versaNucs; homotypic NCPs unless stated otherwise] for *dCypher* were synthesized/purified/assembled as previously (*Shah et al., 2018*; *Thålin et al., 2020*) but without DNA barcoding (*Supplementary file 2C–D*; and *Figure 1—figure supplement 2*). PTMs were confirmed by mass-spectrometry and immunoblotting (if an antibody was available) (*Weinberg et al., 2019*; *Goswami et al., 2021*; *Marunde et al., 2022b*).

For dNucs, PTM-defined histones were mixed (at mg scale) to a defined stoichiometry and dialyzed/purified to octamers, which were subsequently assembled on 147 bp 5' biotinylated 601 DNA (*Dyer et al., 2003*). The resulting products (e.g. H3K4me3; *EpiCypher#16–0316*) contained full-length 'scarless' histones and minimal free-DNA (<5%).

For versaNucs, histone H3 tail peptides (aa1-31; A29L) with a PTM (or mutation) of interest were individually ligated to a H3 tailless nucleosome precursor (H3.1NΔ32 assembled on 147 bp 5' biotinylated 601 DNA; *EpiCypher#16–0016*). The resulting nucleosomes (assembled at 50–100 μg scale) contained minimal free DNA (<5%), undetectable levels of peptide precursor, and ≥90% full-length H3.1 with the PTM(s)/mutations of interest (e.g. *Figure 1—figure supplement 2*; *Thålin et al., 2020*). In general, we observed no discernible difference in dNuc and versaNuc behavior (not shown), so they are used interchangeably in this study (while always including both forms if available; *Supplementary file 2C–D*). However, versaNucs are not recommended for studies that encroach on the A29L position (as present in the final product): e.g., for modifiers or binders to H3R26, K27, or S28.

Heterotypic nucleosomes (*Supplementary file 2C*) were created by approaches to be detailed elsewhere (manuscript in preparation). In brief, PTM-defined H3 histones were reacted to create obligate heterodimers joined by an N-terminal bridge containing a proteolytic cleavage site. Bridging was established by a one-way 'click'-like reaction prior to N-terminal peptide ligation to C-terminal histone cores (i.e. yielding only AB; no AA or BB). Covalently bridged H3 heterodimers were assembled to PTM-defined octamers and nucleosomes as for unbridged histones, then enzymatically cleaved to excise the N-terminal bridge and yield an unscarred heterotypic nucleosome. Heterotypic identity was confirmed at all synthesis steps by analyses additional to those used for homotypics, including Nuc-MS on representative final nucleosomes (*Schachner et al., 2021*). Heterotypic nomenclature describes each PTM-defined histone in the NCP, such that [H3K4me3K14ac • H3] vs. [H3K4me3 • H3K14ac] contain the same total PTM complement but distributed *cis* or *trans* on the H3 N-termini.

### *dCypher* binding assays

The *dCypher* approach (*dCypher* for brevity) was developed on the chemiluminescent bead-based, no-wash Alpha platform (*PerkinElmer*) for the high-throughput profiling of CAP binding to PTM-defined histone peptides and semi-synthetic nucleosomes (homotypic NCPs unless stated otherwise) (*Marunde et al., 2022a*; *Morgan et al., 2021*; *Weinberg et al., 2019*; *Dilworth et al., 2022*; *Weinberg et al., 2021*; *Jain et al., 2020*; *Lloyd et al., 2020*). In brief, biotinylated peptides or NCPs (the potential **Targets**) were individually coupled to streptavidin-coated 'Donor' beads, while epitope-tagged proteins (the **Queries**) were bound to anti-tag 'Acceptor' beads. After mixing potential reactants in a 384-well format, Donor beads were excited at 680 nm, releasing singlet oxygen that caused emission (520–620 nm) in proximal (within 200 nm) Acceptor beads; this luminescent signal is directly correlated to interaction/binding affinity. A complete description of the *dCypher* approach is available (*Marunde et al., 2022a*; *Morgan et al., 2021*). *dCypher* binding assays to PTM-defined peptides/NCPs were as previously (*Weinberg et al., 2019*; *Jain et al., 2020*). In brief 5 µl of GST- or 6HIS-tagged reader domain (Query: specific identity/concentration as indicated) was incubated with 5 µl of biotinylated peptide (100 nM final)/NCP (10 nM final) (Target: specific identity as indicated) for 30 min at room temperature in the appropriate assay buffer ([Peptide: 50 mM Tris pH 7.5, 50 mM NaCl, 0.01% Tween-20, 0.01% BSA, 0.0004% Poly-L Lysine, 1 mM TCEP]; [NCP: 20 mM HEPES pH 7.5, 250 mM NaCl, 0.01% BSA, 0.01% NP-40, 1 mM DTT]) in a 384-well plate. For GST-tagged proteins, a 10 µl mix of 2.5 µg/ml glutathione (*PerkinElmer*) and 5 µg/ml streptavidin donor beads (*PerkinElmer*) was prepared in peptide or NCP bead buffer ([Peptide: as assay buffer]; [NCPs: as assay buffer minus DTT]) and added to each well. For 6HIS-tagged proteins, a 10 µl mix of 2.5 µg/ml Ni-NTA acceptor beads (*PerkinElmer*) and 10 µg/ml streptavidin donor beads was used. The plate was incubated at room temperature in subdued lighting for 60 min and the Alpha signal was measured on a *PerkinElmer 2104 EnVision* (680 nm laser excitation, 570 nm emission filter ± 50 nm bandwidth). Each binding interaction was performed in duplicate (*Marunde et al., 2022b*). See below for a comparison of data derived from each epitope tag [GST vs. 6His].

### Calculation of $EC_{50}^{rel}$

Binding curves [Query: Target] were generated using a non-linear 4PL curve fit in Prism 8 (*GraphPad*). To rank [Query: Target] binding we used a four-parameter logistical (4PL) model and reported the resulting data fit as relative $EC_{50}$s ($EC_{50}^{rel}$) (*Sebaugh, 2011*) and 95% confidence intervals (CI95) (*Supplementary file 1* for all from the study). These values are defined as the concentration of Query (e.g. GST-PHD-BD) required to provoke a half-maximal response to Target along a representative dose-response curve (*Beck et al., 2012*). Notably, we report as $EC_{50}^{rel}$ because a stable maximal response (100% ± 5%) control is not included during data generation: as such we cannot ensure saturation. Although these values can be directly compared across Queries to understand relative binding (as they are within this study), they are not treated as an equilibrium dissociation constant ($K_d$). Very specific parameters must be met within the set-up of an Alpha assay to define a binding interaction $K_d$: namely the generation of saturation curves, or a competition assay to identify the Query concentration at least 5 x below bead binding saturation using 10 x Target (*Cassel et al., 2010*). Where necessary, values beyond the Alpha hook point (indicating bead saturation/competition with unbound Query) (*Marunde et al., 2022b*) were excluded and top signal constrained to average max signal for Target (in cases where signal never reached plateau, those were constrained to the average max signal within the assay). For statistical analysis, unpaired two-tailed t-tests were performed in Prism using Log($EC_{50}^{rel}$) and standard error values/differences considered statistically significant when $p < 0.05$ (*Supplementary file 1*).

### Comparison of epitope tags [GST vs. 6His]

For *dCypher* assays BPTF PHD-BD and PHD Queries were N-terminally tagged with either GST- or 6His-, while the BD Query was only available as 6His- (due to expression/purification difficulties).

A comparison of both epitope-tagged forms of PHD-BD revealed the resulting $EC_{50}^{rel}$ data for most analogous Targets to rank order identically (*Supplementary file 1*). One interesting exception was for PHD-BD binding to H3K4me1, me2, and me3 peptides. Here, GST-PHD-BD showed similar $EC_{50}^{rel}$ for each H3K4 methyl state, while 6His-PHD-BD displayed a moderate preference for H3K4me3. Furthermore, the $EC_{50}^{rel}$ values of GST-tagged Queries were reduced compared to their 6His-tagged

counterparts, indicating tighter binding. These results may be due to the use of epitope-specific beads (i.e. glutathione vs. nickel chelate acceptor beads; with the former being potentially more sensitive), and/or the dimerization of GST (*Bell et al., 2013*), which would be expected to enhance [Query: Target] binding via an effective local increase in Query concentration. Given the above, we note the importance to only compare $EC_{50}^{rel}$ values between similarly tagged Queries.

## CUTANA CUT&RUN, Illumina sequencing, and data analysis

CUT&RUN was performed with K562 cells (fixed (H3K18ac) or native (all other targets)) using CUTANA protocol v1.5.1 *Yusufova et al., 2021*; an optimized version of that previously described (*Skene et al., 2018*). For each native CUT&RUN reaction, 500 K digitonin permeabilized cells were immobilized to Concanavalin-A beads (Con-A; *EpiCypher* #21–1401) and incubated overnight (4 °C with gentle rocking) with 0.5 µg of antibody (IgG, anti-H3K4me3 or anti-BPTF *Supplementary file 2E*; all PTM antibodies validated to SNAP-ChIP nucleosome standards as previously *Shah et al., 2018*). pAG-MNase (*EpiCypher* #15–1016) was added/activated and CUT&RUN enriched DNA was purified using the *Monarch DNA Cleanup* kit (*NEB* #T1030S). 10 ng DNA was used to prepare sequencing libraries with the *Ultra II DNA Library Prep* kit (*NEB* #E7645S).

Some labile PTMs benefit from a light fixation step (not shown), so minor protocol modifications were made for H3K18ac. 500 K cells were crosslinked with 0.1% formaldehyde for 1 min at room temperature, and then quenched with 125 mM glycine. To help the cellular ingress of antibody/ egress of cleaved chromatin fragments the Wash, Antibody, and Digitonin buffers were supplemented with 1% Triton X-100 and 0.05% SDS. To reverse crosslinks prior to DNA column cleanup, CUT&RUN eluate was incubated overnight at 55 °C with 0.8 µl 10% SDS and 20 µg Proteinase K (*Ambion* #AM2546).

Libraries were sequenced on the Illumina platform, obtaining ~4 million paired-end reads on average (*Supplementary file 2E*). Paired-end fastq files were aligned to the hg19 reference genome using Bowtie2 (*Langmead and Salzberg, 2012*). Uniquely aligned reads were retained, and blacklist regions (*Amemiya et al., 2019*) removed before subsequent analyses. Peaks were called using SEACR (Sparse Enrichment Analysis of CUT&RUN) (*Meers et al., 2019*). All sequencing data has been deposited in the NCBI Gene Expression Omnibus (GEO) with accession number GSE150617.

## Reader CUT&RUN

Reader CUT&RUN (i.e. GST-PHD-BD as a detection tool) was performed as above for CUTANA CUT&RUN with the following modifications.

500 K native K562 cells were used for each reaction and all buffers were supplemented with 1 µM TSA (Trichostatin A, *Sigma* #T8552) to protect potentially labile acetyl-PTMs (e.g. H3K18ac).

A biotinylated CUTANA nucleosome mini-panel (unmodified, H3K4me3, H3tetra$^{ac}$, H3K4me3tri$^{ac}$; each on 80-N-25 DNA containing a central 147 bp Widom 601 Nucleosome positioning sequence with embedded 22 bp DNA barcode: *Figure 6a–b*) was synthesized, individually coupled to magnetic streptavidin beads (*NEB* #S1421S) at saturation, and spiked into each CUT&RUN reaction (final concentration 0.8 nM) with Con-A immobilized cells just prior to antibody addition. Each member of the nucleosome panel was DNA barcoded to define PTM status/monitor comparative release into the CUT&RUN eluate (to be quantified after sequencing). After nucleosome spike-in, GST-PHD-BD, GST (*Supplementary file 2A*) or IgG (*Supplementary file 2E*) were added to parallel reactions (each 70 nM final concentration), and incubated overnight at 4 °C. Samples were washed twice, and the appropriate incubated with 0.5 µg anti-GST (*Supplementary file 2E*) at room temperature for 30 min. Remainder of the assay was performed using standard CUT&RUN protocol and sequenced as above. All sequencing data has been deposited in GEO with accession number GSE150617.

## Acknowledgements

HAF was supported by a National Institutes of Health (NIH) T32 fellowship (2T32GM008365-26A1) through the Center for Biocatalysis and Bioprocessing. This project was supported by The Holden Comprehensive Cancer Center at The University of Iowa and its National Cancer Institute Award (P30CA086862). Work in the Musselman lab was funded by grants from the National Science Foundation (CAREER-1452411) and the NIH (R35GM128705). *EpiCypher* is supported by NIH grants R43CA236474, R44GM117683, R44CA214076, R44GM116584, and R44DE029633. We would like to

thank Vic Parcell and the High-Resolution Mass Spectrometry Facility (Office of the Vice-President for Research and Economic Development at the University of Iowa) for technical support.

## Additional information

### Competing interests

Matthew R Marunde, Jonathan M Burg, Irina K Popova, Anup Vaidya, Nathan W Hall, Ellen N Weinzapfel, Matthew J Meiners, Rachel Watson, Zachary B Gillespie, Hailey F Taylor, Laylo Mukhsinova, Ugochi C Onuoha, Sarah A Howard, Katherine Novitzky, Eileen T McAnarney, Krzysztof Krajewski, Martis W Cowles, Marcus A Cheek, Zu-Wen Sun, Bryan J Venters: Affiliated with EpiCypher Inc; the author has no financial interests to declare. Michael-C Keogh: A board member of EpiCypher Inc; the author has no financial interests to declare. The other authors declare that no competing interests exist.

### Funding

| Funder | Grant reference number | Author |
|---|---|---|
| National Institutes of Health | 2T32GM008365-26A1 | Harrison A Fuchs |
| National Institutes of Health | P30CA086862 | Catherine A Musselman |
| National Science Foundation | CAREER-1452411 | Harrison A Fuchs |
| National Institutes of Health | R35GM128705 | Harrison A Fuchs Catherine A Musselman |
| National Institutes of Health | R43CA236474 | Matthew R Marunde |
| National Institutes of Health | R44GM117683 | Matthew R Marunde |
| National Institutes of Health | R44CA214076 | Matthew R Marunde |
| National Institutes of Health | R44GM116584 | Matthew R Marunde |
| National Institutes of Health | R44DE029633 | Matthew R Marunde |

The funders had no role in study design, data collection and interpretation, or the decision to submit the work for publication.

### Author contributions

Matthew R Marunde, Harrison A Fuchs, Conceptualization, Data curation, Formal analysis, Investigation, Writing – original draft, Writing – review and editing; Jonathan M Burg, Resources, Data curation, Formal analysis, Investigation; Irina K Popova, Anup Vaidya, Nathan W Hall, Katherine Novitzky, Data curation, Formal analysis, Investigation; Ellen N Weinzapfel, Martis W Cowles, Project administration; Matthew J Meiners, Rachel Watson, Sarah A Howard, Marcus A Cheek, Resources, Validation; Zachary B Gillespie, Hailey F Taylor, Laylo Mukhsinova, Ugochi C Onuoha, Eileen T McAnarney, Krzysztof Krajewski, Methodology; Zu-Wen Sun, Supervision, Methodology, Writing – review and editing; Bryan J Venters, Data curation, Formal analysis, Investigation, Writing – original draft, Writing – review and editing; Michael-C Keogh, Conceptualization, Formal analysis, Supervision, Funding acquisition, Writing – original draft, Writing – review and editing; Catherine A Musselman, Data curation, Formal analysis, Supervision, Funding acquisition, Investigation, Writing – original draft, Writing – review and editing

### Author ORCIDs

Matthew R Marunde ⓘ https://orcid.org/0009-0007-5934-7200
Eileen T McAnarney ⓘ https://orcid.org/0000-0003-2337-2889

Krzysztof Krajewski ⓘ https://orcid.org/0000-0001-7159-617X
Michael-C Keogh ⓘ https://orcid.org/0000-0002-2219-8623
Catherine A Musselman ⓘ http://orcid.org/0000-0002-8356-7971

**Decision letter and Author response**

Decision letter https://doi.org/10.7554/eLife.78866.sa1
Author response https://doi.org/10.7554/eLife.78866.sa2

---

## Additional files

**Supplementary files**

• Supplementary file 1. *dCypher* data. (A): *dCypher* data analysis EC50rel (lower CI95 - upper CI95). (B) *dCypher* analysis select EC50rel comparison. (C) *dCypher* analysis t-tests. (D) *Figure 1C* raw data (GST-PHD-BD; peptides). (E) *Figure 1D* raw data (GST-PHD-BD; nucleosomes). (F) *Figure 2B* upper raw data (GST-PHD-BD N3007A; nucleosomes). (G) *Figure 2B* lower raw data (GST-PHD-BD W2891A; nucleosomes). (H) *Figure 3B* raw data 6His-PHD-BD; nucleosomes. (I) *Figure 5A* raw data (GST-PHD-BD; heterotypic nucleosomes). (J) Extended *Figure 1B* raw data (GST-PHD-BD; peptide screen). (K) Extended *Figure 1C* raw data (GST-PHD-BD; nucleosome screen). (L) Extended *Figure 1D* raw data (GST-PHD-BD; peptides). (M) Extended *Figure 1E* raw data (GST-PHD-BD; nucleosomes). (N) Extened *Figure 1F* raw data (GST-PHD-BD; nucleosomes). (O) Extended *Figure 2C* raw data (6His-PHD; peptides). (P) Extended *Figure 2D* raw data (6His-PHD; nucleosomes).(Q) Extended *Figure 2E* raw data (GST-PHD; nucleosomes). (R) Extended *Figure 2F* raw data (6His-BD; peptides). (S) Extended *Figure 2G* raw data (6His-BD; nucleosomes). (T) Extended *Figure 3D* raw data (Anti-H3K4me3; nucleosomes). (U) Extended *Figure 4A* raw data (GST-BRD4 BD; peptides). (V) Extended *Figure 4B* raw data (GST-BRD4 BD; nucleosomes)

• Supplementary file 2. Resources. (A) Plasmid nd proteins. (B) *dCypher* peptides (287 x). (C) *dCypher* dNucs (59 x) (incl. heterotypics: 5 x). (D) versaNuc peptides (18 x). (E) CUT&RUN antibodies (and sequence stats).

• MDAR checklist

### Data availability

Raw data from *dCypher* assays is in **Supplementary file 1**. All sequencing data has been deposited in the NCBI Gene Expression Omnibus (GEO) with accession number GSE150617.

The following previously published dataset was used:

| Author(s) | Year | Dataset title | Dataset URL | Database and Identifier |
|---|---|---|---|---|
| Venters BJ | 2022 | Nucleosome conformation dictates the histone code | https://www.ncbi.nlm.nih.gov/geo/query/acc.cgi?acc=GSE150617 | NCBI Gene Expression Omnibus, GSE150617 |

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
