## [Editor Report]

The manuscript investigates how the tandem reader domains in BPTF co-recognize two types of modifications present on histone tails, H3K4me3 and H3 acetylation. The authors provide compelling evidence for regulation of such recognition by the conformational restriction of histone tails due to interactions with nucleosomal DNA. The findings contribute valuable new insights into how the nucleosomal context impacts the action of tandem reader domains and should be of much interest to the broader chromatin field.

---

## [Decision Letter]

**Decision letter after peer review:**

Thank you for submitting your article "Nucleosome conformation dictates the histone code" for consideration by *eLife*. Your article has been reviewed by 2 peer reviewers, and the evaluation has been overseen by a Reviewing Editor and Jessica Tyler as the Senior Editor. The reviewers have opted to remain anonymous.

The reviewers have discussed their reviews with one another and the reviewing editor. Overall the reviewers judged that the work contributes new insights into how multiple histone modifications are recognized by tandem reader domains in a nucleosomal context. However, they also agreed that key in vitro experiments were missing to support the core conclusions about how the BPTF protein binds two separate marks in a nucleosomal context. They have in addition recommendations for more clearly acknowledging prior work and alternative models. Based on these discussions the Reviewing Editor has drafted the list of essential revisions to help you prepare a revised submission.

Essential revisions:

1. In many places the manuscript is written more generally as if the conclusions apply in all cases (e.g. the title, abstract, and introduction) and this remains to be determined. It is also overstated that there is a belief that peptides perfectly recapitulate nucleosomes. Authors should also point out that the nucleosomes are multi-valent and the data cannot discriminate binding of a single PHD-BD to single or multiple tails, and that the work is limited as it is using a construct of BPTF and in fact, there is at least one other reader domain involved.

2. The authors seem to have overlooked the fact that mononucleosome substrates have been in use for determining the substrate specificity and mechanisms of quite a few enzymes that simply do not act on peptide substrates. For example, Dot1L doesn't do anything with peptides nor does COMPASS/Set1, both of which require intact nucleosomal substrates to measure their activity in response to ubiquitylated H2B. Thus, the authors' refinement of the "histone code hypothesis" is unnecessary and overdone. We suggest that they instead cite examples where nucleosome substrates have provided answers that cannot be obtained from peptide substrates alone. For example, extensive work from the Muir and Allis labs.

3. The authors' focus on H3K4 and H3acetylation is informative but a bit puzzling as prior work by Ruthenburg and Allis (reference 33) conducted experimentation with nucleosomes and concluded that H3K4me3-H4K16ac is a modification state-bound preferentially by BPTF in vitro and in cells. And work in reference 34 also shows that acetylation on H4 is co-recognized with methylation on H3K4. The authors have ignored these previous results in their own discussions. Readers would benefit from a side-by-side comparison of the two acetylation states to get a sense of which is a stronger interaction and why both seemingly correlate in CUTnRUN or ChIP-seq. Importantly the authors need to measure the affinity of their GST-PHD-BD construct for nucleosomes containing H3K4me3-H4K16ac and compare it to the affinity for nucleosomes containing H3K4me3K18ac (or H3K4me3triac). The data should then be interpreted in the context of the prior work in references 33 and 34. It is possible that the authors will discover interesting differences in how acetylation is recognized by the BPTF BD on H3 vs H4 tails when it is anchored to the H3K4me3 mark on nucleosomes.

4. The NMR experiments are all undertaken with 150mM KCl with no NaCl present. While NMR experimental constraints are understandable, the authors should avoid sweeping statements from NMR experiments regarding the dynamism of histone tails in chromatin, unless specific experiments are cited/conducted to demonstrate the same in cells. Many factors may contribute to the exclusion of BPTF from modified histone tails in cells, including the binding of other reader proteins, and the precise genomic localization of these modifications vis-a-vis BPTF. The important role of anchoring proteins must also be taken into account when considering binding/non-binding of substrates by CAPs. Thus, the NMR experiments presented in the manuscript do not report on whether BPTF binds H4K16ac in cells or indeed in vitro. If the PHD domain is capable of ultimately binding the H3 tail despite the tail's fuzzy interaction with DNA, the question remains as to why the bromodomain may not do so for acetylated H4 tails? This question also necessitates the in vitro experiment described in point 3 above.

5. The peptide binding studies need some clarification. The work in reference 28 indicates the following Kds for the PHD domain of BPTF: ~2.7 µM for H3K4me3 peptide and 5.0 µM for H3K4me2 peptide. In contrast, the authors here see Kds in the nM range for H3K4me3 for the GST-PHD domain (42 nM EC50). Please clarify why these values are so different. Additionally, if the BD and PHD are indeed capable of co-recognizing H3K4 me3 and H3 K18 Ac, it is puzzling why in the context of the peptides, H3K4me3, H3K4me3tri-ac and H3tetratac are bound with similar affinities (0.88-2.4 nm), with only about a 2-fold enhancement in affinity from having both marks. These data suggest that perhaps these two marks are not bound in cis but rather in trans across different tails. The authors should comment/clarify this issue.

6. Please double-check all figure references. It may have been how the paper was collated during submission, but it was difficult to know which figure/figure supplement was being referred to.

7. Pg 5 line 135, the data shows that there is a stronger affinity for H3K4me3triac. Do the authors mean that PHD-BD preferentially reads out K14ac and K18ac over K9ac in the H3K4me3triac? Clarify.

8. Pg 2 Lines 57-58, Unclear wording, "same" what?

9. Pg 2 line 47, NCP as abbreviation instead as it is more standard.

10. Figure 1A, Include residue numbers on the domain diagram.

---

## [Author Response]

Essential revisions:1. In many places the manuscript is written more generally as if the conclusions apply in all cases (e.g. the title, abstract, and introduction) and this remains to be determined. It is also overstated that there is a belief that peptides perfectly recapitulate nucleosomes. Authors should also point out that the nucleosomes are multi-valent and the data cannot discriminate binding of a single PHD-BD to single or multiple tails, and that the work is limited as it is using a construct of BPTF and in fact, there is at least one other reader domain involved.

We appreciate this view and the text has been altered throughout, including the field approach to peptides, so as not to overstate the generalizability of our results. We also note that BPTF (and indeed NURF) contains more nucleosome binding potential than the PHD-BD tandem that is the focus of this study.

Regarding the multivalent nature of nucleosomes: our revised manuscript includes exciting new data with heterotypics, containing distinct PTM-defined forms of histone H3 within the same nucleosome. In binding studies we demonstrate that BPTF PHD-BD most effectively associates with H3K4me4 and H3K14ac in *cis* vs. *trans* (i.e. [H3K4me3K14ac • H3] vs. [H3K4me3 • H3K14ac] new Figure 5), an important question as noted.

We agree with the reviewer that in vitro experiments can be limited (in this case since we focused on two of the reader domains within BPTF). However, the CUT&RUN (genomic) experiments were carried out with both the PHD-BD [*aka*. exogenous] and full-length [*aka*. endogenous] BPTF. These results show the tandem PHD-BD plays a central role in targeting BPTF in the chromatin context, but also reveal that additional regulation is most likely involved, which may be due to other domains within full-length BPTF (or associated proteins within the NURF complex: see Discussion).

2. The authors seem to have overlooked the fact that mononucleosome substrates have been in use for determining the substrate specificity and mechanisms of quite a few enzymes that simply do not act on peptide substrates. For example, Dot1L doesn't do anything with peptides nor does COMPASS/Set1, both of which require intact nucleosomal substrates to measure their activity in response to ubiquitylated H2B. Thus, the authors' refinement of the "histone code hypothesis" is unnecessary and overdone. We suggest that they instead cite examples where nucleosome substrates have provided answers that cannot be obtained from peptide substrates alone. For example, extensive work from the Muir and Allis labs.

Through the text we have cited studies in which enzyme and reader activity is unique in the nucleosome context, as well as additional cases where nucleosome substrates have proven insightful (see revised citations 20-31). However, we agree we did not draw a clear enough distinction with these studies, and have clarified in the revised text that our approach was unique in that it did not bias the studied nucleosome substrates according to initial peptide results.

3. The authors' focus on H3K4 and H3acetylation is informative but a bit puzzling as prior work by Ruthenburg and Allis (reference 33) conducted experimentation with nucleosomes and concluded that H3K4me3-H4K16ac is a modification state-bound preferentially by BPTF in vitro and in cells. And work in reference 34 also shows that acetylation on H4 is co-recognized with methylation on H3K4. The authors have ignored these previous results in their own discussions. Readers would benefit from a side-by-side comparison of the two acetylation states to get a sense of which is a stronger interaction and why both seemingly correlate in CUTnRUN or ChIP-seq. Importantly the authors need to measure the affinity of their GST-PHD-BD construct for nucleosomes containing H3K4me3-H4K16ac and compare it to the affinity for nucleosomes containing H3K4me3K18ac (or H3K4me3triac). The data should then be interpreted in the context of the prior work in references 33 and 34. It is possible that the authors will discover interesting differences in how acetylation is recognized by the BPTF BD on H3 vs H4 tails when it is anchored to the H3K4me3 mark on nucleosomes.

We fully appreciate prior work by the Allis and Muir groups. We also do not mean to imply that other patterns of modifications cannot be engaged by BPTF, including H3K4me3-H4K16ac (a pattern predicted from peptide studies). Here we did not bias our nucleosome library by peptide data, and this revealed some patterns of PTM readout are only seen in the nucleosome context [*e.g.* Figure 1]. While we appreciate the request to carry out the requested side-by-side comparisons, this would entail >$20k for additional combinatorial nucleosome generation, and is not the major point of our study. Indeed we revised the Discussion to better reflect this point. We also note exciting recent data from a parallel study (led by *EpiCypher* and the Strahl group) showing H3K4me3 is invariably seen in (and most effectively created in) the *cis* context of H3 tail acetylation (doi.org/10.7554/*eLife*.82596). As such the combinatorial H3 binding target we have identified for BPTF PHD-BD is also a preferred PTM pattern created in vivo.

4. The NMR experiments are all undertaken with 150mM KCl with no NaCl present. While NMR experimental constraints are understandable, the authors should avoid sweeping statements from NMR experiments regarding the dynamism of histone tails in chromatin, unless specific experiments are cited/conducted to demonstrate the same in cells. Many factors may contribute to the exclusion of BPTF from modified histone tails in cells, including the binding of other reader proteins, and the precise genomic localization of these modifications vis-a-vis BPTF. The important role of anchoring proteins must also be taken into account when considering binding/non-binding of substrates by CAPs. Thus, the NMR experiments presented in the manuscript do not report on whether BPTF binds H4K16ac in cells or indeed in vitro. If the PHD domain is capable of ultimately binding the H3 tail despite the tail's fuzzy interaction with DNA, the question remains as to why the bromodomain may not do so for acetylated H4 tails? This question also necessitates the in vitro experiment described in point 3 above.

While it is notoriously difficult to study, previous reports have indicated the nuclear ionic composition is dominated by KCl (*e.g.* doi.org/10.1073/pnas.48.5.853; doi.org/10.1085/ jgp.67.4.469; doi.org/10.1113/jphysiol.1978.sp012526). Thus, our chosen NMR conditions were not due to experimental constraints, but actually to better mimic the nuclear environment. However, we fully agree this is an incomplete depiction (even at the ionic level), and there is much more to the picture than just histone tail dynamics. This is expanded upon this in the revised Discussion.

We also agree that we have not fully answered outstanding questions about relative H3 and H4 tail dynamics / accessibility. To fully address this is unfortunately beyond the scope of the current study. However, we did carry out the suggested NMR experiments re. H4K16ac nucleosome tail dynamics and BPTF BD engagement, and this data is included in the revised manuscript (Figure 4).

5. The peptide binding studies need some clarification. The work in reference 28 indicates the following Kds for the PHD domain of BPTF: ~2.7 µM for H3K4me3 peptide and 5.0 µM for H3K4me2 peptide. In contrast, the authors here see Kds in the nM range for H3K4me3 for the GST-PHD domain (42 nM EC50). Please clarify why these values are so different. Additionally, if the BD and PHD are indeed capable of co-recognizing H3K4 me3 and H3 K18 Ac, it is puzzling why in the context of the peptides, H3K4me3, H3K4me3tri-ac and H3tetratac are bound with similar affinities (0.88-2.4 nm), with only about a 2-fold enhancement in affinity from having both marks. These data suggest that perhaps these two marks are not bound in cis but rather in trans across different tails. The authors should comment/clarify this issue.

In methods we discuss the relative EC50 (EC_50_^Rel^) reported by Α and how it is not an equilibrium K_d_. This is better referenced in the revised text and we appreciate the reviewers comment re. this information being difficult to find in the original submission.

The weak combinatorial PTM engagement on H3 tail peptides, but strong on nucleosomes is indeed puzzling and a major point of our study (and stresses the importance of not being biased by initial peptide data).

As the reviewer notes, at initial submission we could not determine if the H3 tails were bound in *cis* or *in trans* (since all nucleosomes were homotypic). However, as noted in point 1, have now added exciting new data with heterotypic nucleosomes demonstrating that the preferred engagement is *cis* (*i.e*. [H3K4me3tri^ac^ • H3] > [H3K4me3 • H3tetra^ac^] : new Figure 5).

6. Please double-check all figure references. It may have been how the paper was collated during submission, but it was difficult to know which figure/figure supplement was being referred to.

These have been revised to accommodate all updates and confirmed as accurate.

7. Pg 5 line 135, the data shows that there is a stronger affinity for H3K4me3triac. Do the authors mean that PHD-BD preferentially reads out K14ac and K18ac over K9ac in the H3K4me3triac? Clarify.

We meant in comparison to H3K4me3 alone and it has been clarified in the text.

8. Pg 2 Lines 57-58, Unclear wording, "same" what?

This has been clarified in the revised text.

9. Pg 2 line 47, NCP as abbreviation instead as it is more standard.

We have eliminated the Nuc abbreviation and expanded to nucleosome or changed to NCP when appropriate throughout.

10. Figure 1A, Include residue numbers on the domain diagram.

These have been included.